



# Modelled land use and land cover change emissions - A spatio-temporal comparison of different approaches

Wolfgang A. Obermeier[1], Julia E.M.S. Nabel[2], Tammas Loughran[1], Kerstin Hartung[1,*], Ana Bastos[3], Felix Havermann[1], Peter Anthoni[4], Almut Arneth[4], Daniel S. Goll[5], Sebastian Lienert[6], Danica Lombardozzi[7], Sebastiaan Luyssaert[8], Patrick C. McGuire[9], Joe R. Melton[10], Benjamin Poulter[11], Stephen Sitch[12], Michael O. Sullivan[12], Hanqin Tian[13], Anthony P. Walker[14], Andrew J. Wiltshire[12,15], Soenke Zaehle[4], and Julia Pongratz[1,2]

[1]Department of Geography, Ludwig Maximilians Universität, Luisenstrasse 37, 80333 Munich, Germany
[2]Max Planck Institute for Meteorology, 20146 Hamburg, Germany
[3]Max Planck Institute for Biogeochemistry, 07745 Jena, Germany
[4]Karlsruhe Institute of Technology, Institute of Meteorology and Climate Research / Atmospheric Environmental Research, 82467 Garmisch-Partenkirchen, Germany
[5]Laboratoire des Sciences du Climat et de l'Environnement (LSCE), 91191 Gif-sur-Yvette, France
[6]Climate and Environmental Physics, Physics Institute and Oeschger Centre for Climate Change Research, University of Bern, Bern 3012, Switzerland
[7]National Center for Atmospheric Research (NCAR), Climate & Global Dynamics Lab, Boulder, USA
[8]Department of Ecological Science, Vrije Universiteit Amsterdam, 1081HV Amsterdam, the Netherlands
[9]Department of Meteorology, University of Reading, Earley Gate, Reading RG6 6BB, UK
[10]Climate Processes Section, Climate Research Division, Environment and Climate Change Canada, Victoria, BC, Canada
[11]NASA Goddard Space Flight Center, Biospheric Sciences Laboratory, Greenbelt, Maryland 20771, USA
[12]College of Life and Environmental Sciences, University of Exeter, Exeter EX4 4RJ, UK
[13]School of Forestry and Wildlife Sciences, Auburn University, 602 Ducan Drive, Auburn, AL 36849, USA
[14]Climate Change Science Institute & Environmental Sciences Division, Oak Ridge National Laboratory, Oak Ridge, TN 37831, USA
[15]Met Office Hadley Centre, FitzRoy Road, Exeter EX1 3PB, UK
[*]Now at: German Aerospace Center, Institute of Athmospheric Physics, 82234 Oberpfaffenhofen, Germany

**Correspondence:** Wolfgang A. Obermeier (wolfgang.obermeier@lmu.de)

**Abstract.** Quantifying the net carbon flux from land use and land cover changes ($f_{\mathrm{LULCC}}$) is critical for understanding the global carbon cycle, and hence, to support climate change mitigation. However, large-scale $f_{\mathrm{LULCC}}$ is not directly measurable, but has to be inferred from models instead, such as semi-empirical bookkeeping models, and process-based dynamic global vegetation models (DGVMs). By definition, $f_{\mathrm{LULCC}}$ estimates are not directly comparable between these two different model types. As

an example, DGVM-based $f_{\mathrm{LULCC}}$ in the annual global carbon budgets is estimated under transient environmental forcing and includes the so-called Loss of Additional Sink Capacity (LASC). The LASC accounts for the impact of environmental changes on land carbon storage potential of managed land compared to potential vegetation which is not represented in bookkeeping models. In addition, $f_{\mathrm{LULCC}}$ from transient DGVM simulations differs depending on the arbitrary chosen simulation time period and the historical timing of land use and land cover changes (including different accumulation periods for legacy

effects). An approximation of $f_{\mathrm{LULCC}}$ by DGVMs that is independent of the timing of land use and land cover changes and their legacy effects requires simulations assuming constant pre-industrial or present-day environmental forcings. Here, we





analyze three DGVM-derived $f_{LULCC}$ estimations for twelve models within 18 regions and quantify their differences as well as climate- and $CO_2$-induced components. The three estimations stem from the commonly performed simulation with transiently changing environmental conditions and two simulations that keep environmental conditions fixed, at pre-industrial and present-
day conditions. Averaged across the models, we find a global $f_{LULCC}$ (under transient conditions) of $2.0 \pm 0.6$ PgC yr$^{-1}$ for 2009–2018, of which $\sim$40% are attributable to the LASC ($0.8 \pm 0.3$ PgC yr$^{-1}$). From 1850 onward, $f_{LULCC}$ accumulated to $189 \pm 56$ PgC with $40 \pm 15$ PgC from the LASC. Regional hotspots of high cumulative and annual LASC values are found in the USA, China, Brazil, Equatorial Africa and Southeast Asia, mainly due to deforestation for cropland. Distinct negative LASC estimates, in Europe (early reforestation) and from 2000 onward in the Ukraine (recultivation of post-Soviet abandoned
agricultural land), indicate that $f_{LULCC}$ estimates in these regions are lower in transient DGVM- compared to bookkeeping-approaches. By unraveling spatio-temporal variability in three alternative DGVM-derived $f_{LULCC}$ estimates, our results call for a harmonized attribution of model-derived $f_{LULCC}$. We propose an approach that bridges bookkeeping and DGVM approaches for $f_{LULCC}$ estimation by adopting a mean DGVM-ensemble LASC for a defined reference period.

*Copyright statement.* TEXT

# 1 Introduction

Terrestrial ecosystems play an important role for the global carbon cycle as they act as substantial sinks and sources of carbon (C) (Keenan and Williams, 2018). In both directions, fluxes in the land carbon cycle have significantly been altered in previous centuries due to anthropogenic land use and land cover changes (LULCCs), in particular by deforestation e.g. driven by early agricultural expansion in high-latitudes and more recent tropical deforestation or recent regional reforestation and afforestation
(denoted reforestation in the following) in high-latitudes (Klein Goldewijk et al., 2011). Since 1850, the accumulated global net flux from LULCC ($f_{LULCC}$) contributed approximately by a third to global anthropogenic $CO_2$ emissions and was the dominant source until the 1950s, when fossil fuel emissions drastically increased (Friedlingstein et al., 2019). Despite its decreasing relative contribution, $f_{LULCC}$ comprises an important share of the global carbon budget (GCB) and might again account for the bulk of anthropogenic C emissions in future, if fossil emissions can be drastically reduced as described in
some socio-economic pathways (Popp et al., 2017; Krause et al., 2018). In line, $f_{LULCC}$ may gain an important role in the quest for negative $CO_2$ emissions technologies, with LULCCs such as reforestation bearing significant potential to sequester atmospheric $CO_2$ (Griscom et al., 2017; Fuss et al., 2018; Sonntag et al., 2016; Arneth et al., 2017). Accordingly, $f_{LULCC}$ quantification is essential to better understand global carbon cycle dynamics, to estimate future climate change, and to support the assessment of greenhouse gas reduction efforts (Friedlingstein et al., 2019).
Irrespective of the $f_{LULCC}$ importance, there is so far no general agreement on a single valid definition and approach to assess it. This is because $f_{LULCC}$ cannot be directly measured on global scale due to the co-occurrence with natural C sinks and sources. For example, in managed forests, C fluxes result from logging and subsequent regrowth, which is part of $f_{LULCC}$, but also in





response to interannual variability or long-term trends in environmental conditions (Friedlingstein et al., 2019). Inventories or satellite-based measurements cannot distinguish C fluxes induced by LULCC from those induced by environmental changes.

To separate these terms, models are applied. Here, various approaches exist. In the 2019 GCB of the Global Carbon Project (named GCB2019 in the following; Friedlingstein et al. 2019), two bookkeeping models are used, 'Bookkeeping of Land Use Emissions' (hereafter BLUE; Hansis et al. 2015) and 'Houghton and Nassikas 2017' (hereafter H&N2017; Houghton and Nassikas 2017). The bookkeeping mean $f_{\mathrm{LULCC}}$ in the GCB2019 is combined with the uncertainty derived from process-based dynamic global vegetation models (DGVMs). DGVMs exist in much larger numbers and their process-based methods

to calculate C fluxes allow to account for the interplay of multiple drivers on C fluxes which bookkeeping models cannot. However, estimates from bookkeeping models and DGVMs are not directly comparable due to underlying assumptions on C stocks (Pongratz et al., 2014). Bookkeeping models are semi-empirical models that combine observation-based C densities with information on areas affected by different types of LULCCs and response curves characterizing the speed of C uptake and release after specific LULCCs to calculate $f_{\mathrm{LULCC}}$. In contrast, to isolate the LULCC effects from those of environmental

changes, DGVM-based $f_{\mathrm{LULCC}}$ is generally estimated as the difference of net land C uptake from net biome productivity (NBP) between simulations with and without LULCC. Within the GCB2019, these simulations are conducted under transient environmental conditions (such as climate, $CO_2$ concentrations and nitrogen deposition), therefore, synergistic fluxes between LULCCs and environmental changes are included.

Inevitably, the transient DGVM approach includes the Loss of Additional Sink Capacity (LASC), representing $CO_2$ fluxes in

response to environmental changes on managed land (typically croplands with low C sink capacity and fast turnover rates) as compared to potential natural vegetation (typically forests with large C sink capacity and slower turnover rates; Gitz and Ciais 2003; Pongratz et al. 2014; Gasser and Ciais 2013; Peng et al. 2014). As an example, when an area which acted as C sink is deforested, the stored C is typically emitted representing environmental conditions at harvest time corresponding to an instantaneous $f_{\mathrm{LULCC}}$. The resulting agricultural area typically does not constitute a major sink. In the simulation without

LULCCs, the forest persists and may increase its C density over time, storing additional C in its slow-turnover woody and soil C pools in response to favourable environmental changes such as increased $CO_2$ concentrations. Compared to the simulation without LULCC, the sink capacity would consequently be diminished in the simulation with LULCC. Thus, even after the emissions of the deforestation event may have ceased, deforestation continues to cause fluxes attributed to $f_{\mathrm{LULCC}}$ due to the reference simulation assuming potential vegetation cover in the absence of LULCCs, and its response to environmental

changes. These theoretical emissions via lost C uptake potential due to human Earth system alterations thus capture the foregone sinks a given LULCC event destroys (or creates, e.g. for reforestation), and accumulate even in absence of further LULCC as long as environmental conditions keep changing in the same direction.

The result of a permanent reduction of a C sink on the LASC as due to a conversion described above, is difficult to predict over time. Natural C sinks are subject to changes, and can even turn into C sources for periods of time, due to the interplay of

multiple factors which control the C balance of ecosystems simultaneously. For example, the LASC may increase because of an increased C uptake via higher NBP resulting from atmospheric $CO_2$ increases (Albani et al. 2006; Schimel et al. 2015, or review of $CO_2$ effect in Walker et al. 2020) or global warming induced longer growing seasons in northern latitudes and higher





altitudes (Keenan et al., 2014; O'Sullivan et al., 2020). Conversely, an increased frequency and severity of drought and heat stress events (Bastos et al., 2020) or increased fire (included in some DGVMs of the GCB2019) may reduce NBP and thus

may cause LASC decreases (and lower $f_{\text{LULCC}}$ estimates) if the C stocks of the potential vegetation in the simulation without LULCCs decrease over time. The LASC will thus differ in magnitude and direction over time and across space.

Environmentally induced C stock changes not only alter the LASC, but also the instantaneous $f_{\text{LULCC}}$. For example, $f_{\text{LULCC}}$ from clearing pristine forest is expected to be higher today than during pre-industrial times if the forest has grown denser over time. Additionally, legacy effects result from the ongoing adaption of ecosystems to historical environmental changes

(Krause et al., 2020). Such transient environmental effects are excluded in bookkeeping approaches – either through using constant C densities, or through purposefully excluding alterations in C densities from transient DGVM simulations in reduced-complexity Earth system models (Gasser et al., 2020). The independence or dependence of vegetation and soil C densities from environmental conditions is thus another difference between transient DGVM and bookkeeping approaches. Here, DGVM simulations under constant environmental forcing can help to attribute $f_{\text{LULCC}}$ quantities independent of the timing of LULCCs.

DGVM simulations under constant environmental conditions have been performed within the project 'Trends and drivers of the regional-scale sources and sinks of carbon dioxide' (TRENDY; Le Quéré et al. 2013; Sitch et al. 2015), when conducting the simulations for the GCB2019 (Friedlingstein et al., 2019). This included a first set of simulations that quantify $f_{\text{LULCC}}$ based on constant present-day environmental conditions. This approach is more similar to bookkeeping estimates and can be evaluated against Earth observation or inventory data as it most closely represents the observable state under today's conditions

and excludes transient flux alterations. Moreover, recent observations are commonly used to estimate the past, for example by combining observed C densities with vegetation coverage reconstructions to infer C stocks in human absence, or with historical area changes for time-series of C stock losses (Sanderman et al., 2017; Erb et al., 2018).

However, as $f_{\text{LULCC}}$ quantities derived under constant present-day conditions are independent of the time at which specific LULCCs occur (unaffected by long-term environmental trends; compare Fig. 1 for illustration), the increased C stocks due

to spin-up with present-day environmental conditions may lead to comparably higher $f_{\text{LULCC}}$ estimates, especially in early simulation years (environmental changes during the industrial period, in general and on global scale, increased C stocks). More realistic $f_{\text{LULCC}}$ estimates for the early period can be derived assuming that pre-industrial environmental conditions prevailed over time (Pongratz et al., 2014; Stocker and Joos, 2015), however, despite being based on the same land use data set, this leads to comparably lower $f_{\text{LULCC}}$ estimates in particular for later LULCCs (Stocker and Joos, 2015).

Assuming constant environmental conditions or C densities over time is clearly unrealistic and requires an arbitrary decision on the time period to determine these variables' values. On the other hand, DGVM-based $f_{\text{LULCC}}$ under more realistic, transient environmental conditions does not correspond to observable fluxes. This poses the question about a proper definition of $f_{\text{LULCC}}$ for a robust and realistic attribution which is valid across time and space. In line, it needs to be decided whether the LASC should be included or excluded (as argued e.g. in Gasser and Ciais 2013; Gasser et al. 2020) as part of $f_{\text{LULCC}}$ and consequently

into the natural land C sink. The urgent need to address this question is underlined by the fact that past LULCCs are estimated to have committed a reduction in the potential global C sink of 80–150 PgC by 2100, which depending on the scenario, translates into a share of ~70% of total global $f_{\text{LULCC}}$ (Strassmann et al., 2008).



This study aims to strengthen the basis for a decision on how to define $f_{\text{LULCC}}$, in particular with respect to the ability of different approaches to resolve the LASC, and thus is a guide on the future role of DGVMs in $f_{\text{LULCC}}$ attribution. To this end, we

present analyses concerning the relevance of different assumptions on environmental conditions, for which the recent extended set of TRENDY DGVM simulations was performed. In particular, our study (1) discusses and quantifies three DGVM-derived $f_{\text{LULCC}}$ (under pre-industrial, transient, and present-day environmental conditions) and bookkeeping estimates in conjunction with their inherent differences on global scale, (2) quantifies the temporal evolution of the differences in DGVM-derived $f_{\text{LULCC}}$ estimates for 18 regions, (3) separates between climate- and $CO_2$-induced $f_{\text{LULCC}}$ components as derived by DGVMs

and (4) aims to approach a spatio-temporally homogenized attribution of $f_{\text{LULCC}}$ as derived by models.

## 2    Data and Methods

This study is based on an ensemble of TRENDY v8 models (http://sites.exeter.ac.uk/trendy/) that ran simulations with and without LULCC for the period 1700–2018 (used in the GCB2019 to quantify $f_{\text{LULCC}}$ uncertainty and to estimate the natural terrestrial C sink; Friedlingstein et al. 2019). It is ensured that all models have reached (1) a steady state after spin-up (offset

in global NBP <0.1 PgC yr$^{-1}$ and drift <0.05 PgC yr$^{-1}$ per century), (2) a net land flux over the 1990s within 90% confidence of constraints by global atmospheric and oceanic observations, and (3) $f_{\text{LULCC}}$ as a C source to the atmosphere over the 1990s (Friedlingstein et al., 2019).

### 2.1    Models and simulations

We use twelve TRENDY v8 DGVMs that provide gridded output of NBP with and without LULCCs under both transient (his-

torically observed) and pre-industrial (constant) environmental conditions (called S0, S2, S3, S4 in the TRENDY v8 protocol; compare Table 2), to calculate the LASC on a regional level (see Table 1 for a comparison of relevant processes included in the DGVMs, additional information can be found in Table A1 in Friedlingstein et al. 2019). For eight models that provided simulations under constant present-day environmental forcing (S5, S6), $f_{\text{LULCC}}$ was also calculated under present-day environmental conditions. All TRENDY v8 simulations were started in 1700 after C stocks reached equilibrium with environmental

conditions in the models, to enable reproducible results with minimized initialization effects for the analyzed time period starting 1850. This implies two separate spin-ups, one for simulations conducted under present-day environmental conditions (S5, S6) and one for those starting from or keeping pre-industrial conditions (all others).

The DGVM simulations with observed transient environmental conditions used observation-based temperature, precipitation, and incoming surface radiation data at $0.5 \times 0.5$ degree spatial resolution of the Climatic Research Unit (CRU) and

Japanese Reanalysis (JRA; Friedlingstein et al. 2019; Harris et al. 2014). Annual time series of global atmospheric $CO_2$ concentrations for 1700–2018 was derived from ice core data (before 1958; Joos and Spahni 2008) merged with National Oceanic and Atmospheric Administration (NOAA) data (from 1958 onward; Dlugokencky and Tans 2020). Models used the HYDE land-use change data set which provides annual, half-degree, fractional data on cropland, rangeland and pasture areas based on annual FAO statistics (Klein Goldewijk et al., 2017; Goldewijk et al., 2017) or the updated harmonised land-use change data





**Table 1.** Overview of the TRENDY v8 DGVMs used and of selected processes included relevant for $f_{LULCC}$. Additionally indicated is if a plausible derivation of the Environmental Equilibrium Difference (EED, compare Eq. 6 and Sect. 2.2.1) and 'Present-day' vs 'Transient' environmental conditions Difference (PTD, compare Eq. 8 and Sect. 2.2.1) was possible.

| Model | Reference | Wood harvest & forest degradation | Shifting cultivation & sub-grid-scale transitions | Irriga-tion | N Fer-tilisa-tion | EED & PTD |
|---|---|---|---|---|---|---|
| CLASS-CTEM | Melton and Arora (2016) | no | no | no | no | yes |
| CLM5.0 | Lawrence et al. (2019) | yes | yes | yes | no | no |
| DLEM | Tian et al. (2015) | yes | no | yes | yes | yes |
| JSBACH | Mauritsen et al. (2019) | yes | yes | no | no | yes |
| JULES-ES 1.02 | Sellar et al. (2019) | yes | no | no | yes | no |
| LPJ-GUESS | Smith et al. (2014) | yes | yes | yes | yes | yes |
| LPJ | Poulter et al. (2011) | yes | yes | no | no | no |
| LPX-Bern | Lienert and Joos (2018) | no | no | no | yes | yes |
| OCN | Zaehle et al. (2011) | yes | no | no | yes | no |
| ORCHIDEE | Krinner et al. (2005) | yes | no | no | no | yes |
| ORCHIDEE-CNP | Goll et al. (2017) | no | no | no | yes | yes |
| SDGVM | Walker et al. (2017) | no | no | no | no | yes |

**Table 2.** Overview of the simulations used in our study, comprising transient (observed historical evolution), pre-industrial or present-day (constant) forcing for environmental conditions (such as climate, atmospheric $CO_2$ concentrations and nitrogen deposition), and transient or pre-industrial LULCC (with an additional description of their purpose of use). For the underlying forcing data and protocol, refer to Friedlingstein et al. (2019). All runs were performed within the TRENDY v8 efforts for the GCB2019.

| Simu-lation | Climate | $CO_2$ con-centration | Nitrogen deposition | Nitrogen fertilization | LULCC forcing | Purpose |
|---|---|---|---|---|---|---|
| S0 | pre-ind. | pre-ind. | pre-ind. | pre-ind. | pre-ind. | control to S4 |
| S1 | pre-ind. | observed | observed | pre-ind. | pre-ind. | vs S0: isolation of $CO_2$/Ndepo effects |
| S2 | observed | observed | observed | pre-ind. | pre-ind. | control to S3 |
| | | | | | | vs S1: isolation of climate effects |
| S3 | observed | observed | observed | observed | LUH2/HYDE | S2–S3: $f_{LULCC}$ under transient env. |
| S4 | pre-ind. | pre-ind. | pre-ind. | observed | LUH2/HYDE | S0–S4: $f_{LULCC}$ under pre-ind. env. |
| S5 | pres.-day | pres.-day | pres.-day | observed | LUH2/HYDE | S6–S5: $f_{LULCC}$ under pres.-day env. |
| S6 | pres.-day | pres.-day | pres.-day | pre-ind. | pre-ind. | control to S5 |



(LUH2; Hurtt et al. 2011, 2020). While HYDE agricultural areas are used in LUH2, the main difference lies in LUH2 addition-
ally adding wood harvest from the Global Forest Resources Assessments of the FAO and sub-grid-scale ('gross') transitions to
capture shifting cultivation in the tropics.

    For pre-industrial simulations, the $CO_2$ concentration and LULCC data from 1700, and nitrogen fertilization and deposition
data from 1860 (no earlier data available) were applied. Climate was derived by recycling the mean and variability from 1901–

1920. For present-day simulations, the $CO_2$ concentration from 2018 and average nitrogen deposition from 1999–2018 were
taken constant, and climate was derived by recycling the mean and variability from 1999–2018.

## 2.2   Data processing

### 2.2.1   Three alternative $f_{\text{LULCC}}$ estimates and their differences

We estimate three different DGVM-based $f_{\text{LULCC}}$s as differences in NBP of a simulation with and one without LULCCs (com-

pare Eq. 1 to 3). Using yearly aggregated NBP values, $f_{\text{LULCC}}$ is derived for each DGVM, time step and grid cell under transient
(subscript $trans$), constant pre-industrial ($pi$), and constant present-day ($pd$) environmental conditions from the TRENDY v8
simulations as follows:

$$f_{\text{LULCC\_trans}} = NBP_{\text{S2}} - NBP_{\text{S3}} \tag{Eq. 1}$$

$$f_{\text{LULCC\_pi}} = NBP_{\text{S0}} - NBP_{\text{S4}} \tag{Eq. 2}$$

$f_{\text{LULCC\_pd}} = NBP_{\text{S6}} - NBP_{\text{S5}}$            (Eq. 3)

    A lower NBP in the simulation including LULCCs compared to the one excluding LULCCs (control) represents a net flux
of $CO_2$ out of the terrestrial biosphere into the atmosphere (emissions) due to LULCCs causing C losses. Conversely, a higher
NBP in the simulation including LULCCs relates to a net flux from the atmosphere into the biosphere due to LULCCs that
enhanced C uptake.

As outlined in the introduction, the derivation of $f_{\text{LULCC\_trans}}$ (Eq. 1; definition as used for uncertainty assessment in the GCB;
Friedlingstein et al. 2019) inherently includes the LASC. The LASC represents theoretical emissions resulting from transient
alterations of environmental conditions since the beginning of the simulation runs (historical changes in climate, atmospheric
$CO_2$ and N deposition, the latter for models including N-cycling), and thus, can be quantified with reference to $f_{\text{LULCC\_pi}}$, fluxes
which would have occurred if pre-industrial environmental conditions prevailed during and after the time LULCCs occurred

(Eq. 4; e.g. Strassmann et al. 2008; Pongratz et al. 2009; Gitz and Ciais 2003; Gasser et al. 2020).





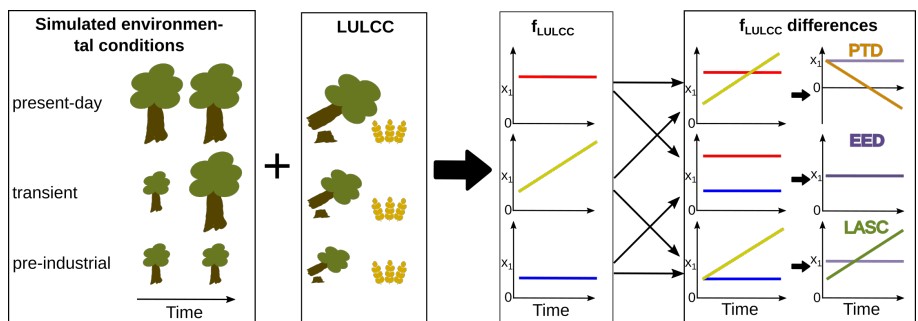

**Figure 1.** Illustration of the different $f_{\text{LULCC}}$ estimations and their differences. The altered sizes of trees (box 1) indicate that vegetation responds to the historical trends in environmental conditions (such as increased $CO_2$ levels and global warming). Historically and globally environmental changes led to an increase in land C stocks, therefore present-day environmental conditions are associated with taller trees in our scheme. When a LULCC occurs that reduces C stocks (box 2) the higher C stocks will cause a higher $f_{\text{LULCC}}$ (box 3: red line higher than blue line; yellow line increasing with time). $f_{\text{LULCC}}$ is derived by subtracting net biome productivity from a simulation without LULCCs from one with LULCCs. Additionally, the different $f_{\text{LULCC}}$ estimations can be compared to each other (box 4): the Loss of Additional Sink Capacity (LASC; compare Eq. 4), Environmental Equilibrium Difference (EED, compare Eq. 6) and 'Present-day' vs 'Transient' environmental conditions Difference (PTD, compare Eq. 8).

$$LASC = f_{\text{LULCC\_trans}} - f_{\text{LULCC\_pi}} \qquad \text{(Eq. 4)}$$

$$= (NBP_{\text{S2}} - NBP_{\text{S3}}) - (NBP_{\text{S0}} - NBP_{\text{S4}}) \qquad \text{(Eq. 5)}$$

The LASC hinders comparison of $f_{\text{LULCC\_trans}}$ with flux estimates based on present-day environmental conditions ($f_{\text{LULCC\_pd}}$). Per definition, the latter represent the closest approximation of bookkeeping fluxes and recent C density observations via

DGVMs. Therefore, we compare $f_{\text{LULCC\_trans}}$ and $f_{\text{LULCC\_pd}}$ to determine times and regions that are most sensitive to the differences introduced when DGVM-derived $f_{\text{LULCC\_trans}}$ is jointly used with bookkeeping estimates, as in the GCB. We call this the 'Present-day' vs 'Transient' environmental conditions Difference (PTD) and derive it according to Eq. 6:

$$PTD = f_{\text{LULCC\_pd}} - f_{\text{LULCC\_trans}} \qquad \text{(Eq. 6)}$$

$$= (NBP_{\text{S6}} - NBP_{\text{S5}}) - (NBP_{\text{S2}} - NBP_{\text{S3}}) \qquad \text{(Eq. 7)}$$

It is not clear even at global scale if PTD is negative or positive. On the one hand, $f_{\text{LULCC\_pd}}$ can be higher than $f_{\text{LULCC\_trans}}$ because C stocks had been brought into equilibrium with present-day conditions during spin-up, i.e. ecosystems had time to equilibrate with high $CO_2$ levels, implying more biomass and higher soil C stocks being affected by – historically prevalent – deforestation. On the other hand, the LASC accumulates over time (Sect. 1 and Fig. 1 for illustration) and therefore $f_{\text{LULCC\_trans}}$ could become larger than $f_{\text{LULCC\_pd}}$. This difference is assumed to be particularly pronounced in former forested areas under



beneficial environmental conditions over the past where LULCCs happened early, as here the LASC could accumulate for a long time (high sensitivity of forest productivity to rising $CO_2$ in DGVMs, compare e.g. Peng et al. 2014).

LASC and PTD add up to the difference of $f_{\text{LULCC\_pd}}$ and $f_{\text{LULCC\_pi}}$. The latter two are derived under constant environmental forcing, meaning that both are indifferent to the timing of LULCCs and their legacy effects (compare Fig. 1 for illustration). However, the choice of the time period from which constant environmental conditions are taken is arbitrary. Nonetheless,

comparison of these two simulations is interesting, as they span the minimum and maximum range of assumptions on environmental conditions that would make sense to consider under typical industrial-era simulations. Up to now, no comparison of $f_{\text{LULCC\_pi}}$ with $f_{\text{LULCC\_pd}}$ exists in the literature, which is why we derive their difference and introduce it as the Environmental Equilibrium Difference (EED; compare Eq. 8).

$$EED = f_{\text{LULCC\_pd}} - f_{\text{LULCC\_pi}} \tag{Eq. 8}$$

$$= (NBP_{\text{S6}} - NBP_{\text{S5}}) - (NBP_{\text{S0}} - NBP_{\text{S4}}) \tag{Eq. 9}$$

Twelve TRENDY v8 DGVMs were compared regarding $f_{\text{LULCC\_pi}}$, $f_{\text{LULCC\_trans}}$ and LASC. $f_{\text{LULCC\_pd}}$ (consequently also EED and PTD) could not be derived for CLM5.0, JULES, LPJ and OCN (no S5 and S6 simulation; eight models). A discussion on the performance of individual models can be found in the appendix section A1.

To get an insight into the spatial trends and drivers of the three DGVM-derived $f_{\text{LULCC}}$ estimates and their differences, a

regional analysis was conducted based on the RECCAP2 regions defined in Tian et al. (2019) and shown in Fig. A2. Since all global and regional analyses were performed based on the original model output, the RECCAP2 map was regridded to each model's native resolution using largest area fraction remapping (to compare globally summed NBP in this study and in the GCB2019, refer to Supplementary Fig. A10). Note, for grid point-wise comparison, all model output was regridded to $720 \times 360$ grid boxes using first-order conservative remapping (Jones, 1999).

Due to high interannual NBP variability, the resulting regional and global $f_{\text{LULCC}}$ estimates were smoothed by a Savitzky–Golay filter using $5\%$ of the spatially summed annual data points (16 years). Savitzky–Golay smoothing was applied to preserve peak heights and widths which are known to be removed by other smoothing practices such as moving averages.

All data pre-processing and statistical analysis was performed using Climate Data Operator software (CDO, v1.9.3; Schulzweida 2019), netCDF Operators (NCO, v4.7.7; Rew et al. 1997), and raster- (v2.8-4; Hijmans and van Etten 2014), ncdf4- (v1.16.1;

Pierce 2019), matrixStats- (v0.56.0; Bengtsson et al. 2020), and pracma- (v2.2.9; Borchers 2019) packages of the CRAN R universe (v3.4.4; R Core Team 2018).

### 2.2.2  Relative climate- vs $CO_2$-induced $f_{\text{LULCC}}$ components

Climate change-related environmental alterations might increase or decrease NBP over time (compare Sect. 1), and thus, cause higher or lower $f_{\text{LULCC\_trans}}$ and $f_{\text{LULCC\_pd}}$ compared to $f_{\text{LULCC\_pi}}$ or bookkeeping estimates. While increasing $CO_2$

concentrations are assumed to generally increase C stocks across the globe, alterations by other environmental changes (mainly precipitation- and temperature-related) are more heterogeneous. To gain knowledge about the underlying environ-





mental drivers, this study aims to separate between climate- and $CO_2$-induced components of $f_{\text{LULCC}}$. We approximate them using S1 and S2 simulations, which differ only with respect to inclusion of climatic changes (Table 2). Assuming that the proportions of climate- versus $CO_2$-induced C stocks changes (we use the total C stocks in vegetation and soil, cTot) translate

linearly into the $CO_2$-induced $f_{\text{LULCC\_trans}}$ component at each grid cell ($f_{\text{LULCC\_CO}_2}$), we derive the latter based on the ratio of cTot in S1 to S2 simulations (Eq. 10). The validity of this approach is supported by $f_{\text{LULCC}}$ in many regions correlating well with biomass stocks across models (Li et al., 2017). Thus, although LULCCs may affect C stocks with different strengths – based on the extent, practice and local ecosystem conditions (including C stock distribution) – it seems appropriate to assume that $f_{\text{LULCC}}$ is not independent from the environmental driver of C stock changes.

$$f_{\text{LULCC\_CO}_2} = f_{\text{LULCC\_trans}} \times (\text{cTot}_{S1}/\text{cTot}_{S2}) \qquad\qquad\qquad\qquad\qquad\qquad (\text{Eq. 10})$$

Ratios of cTot were derived based on the annual averages in the last decade of the simulation period across all models (2009–2018). Due to generally increased differences and ratios of $\text{cTot}_{S1}$ and $\text{cTot}_{S2}$ over the simulated period (compare Fig. A1), our $f_{\text{LULCC\_CO}_2}$ provides the maximum possible contribution of $CO_2$-induced change in $f_{\text{LULCC}}$. C stocks from LPX-Bern and CLM5.0 were excluded from derivation of multi-model mean C stocks due to very high values in particular in high latitudes of

the Northern Hemisphere due to inclusion of peatlands (for LPX-Bern, compare Spahni et al. 2013). C stock outliers smaller than zero were excluded.

As no TRENDY v8 control simulation with pre-industrial LULCC and $CO_2$ concentrations and observed (transient) climate exists, we indirectly assess the climate-only $f_{\text{LULCC}}$ component ($f_{\text{LULCC\_Climate}}$; Eq. 11). Synergies between effects of $CO_2$ concentrations and climatic changes on $f_{\text{LULCC}}$ in the DGVMs are assumed zero in this case. While in reality they may be

substantial (e.g. increased water use efficiency due to stomatal closure under elevated $CO_2$), it is beyond the possibilities of available data to quantitatively assess these synergistic effects.

$$f_{\text{LULCC\_Climate}} = f_{\text{LULCC\_Climate}} - f_{\text{LULCC\_CO}_2} \qquad\qquad\qquad\qquad\qquad (\text{Eq. 11})$$
$$= f_{\text{LULCC\_trans}} - f_{\text{LULCC\_trans}} \times (\text{cTot}_{S1}/\text{cTot}_{S2}) \qquad\qquad (\text{Eq. 12})$$

Note, this climate impact roughly represents the trend in the last hundred years as pre-industrial and present-day climate

conditions are the recycled climate in the earliest decades of the 20th and 21st century, respectively.

## 3    Results and discussion

### 3.1    Differences in $f_{\text{LULCC}}$ estimates on global scale

A general overview of most recent estimates of $f_{\text{LULCC}}$ shows that our estimates are in good agreement to the published ones (Friedlingstein et al. 2019; Gasser et al. 2020; Tables 3 to 5). Slight differences (<0.1 PgC yr$^{-1}$) between $f_{\text{LULCC\_trans}}$ derived

in this study and the DGVM-derived GCB2019 estimates are attributable to the fact that we used only a subset ($n = 12$) of





the models analyzed within the GCB2019 ($n = 15$), to consistently use the same models for the flux and bias estimates on a spatio-temporal level, where possible. The LASC explains the relatively high difference of $f_{LULCC\_trans}$ to the bookkeeping estimates in the GCB2019 and by Gasser et al. (2020), since bookkeping models, by their nature, do not include the LASC. Lower LASC estimates in the GCB2019 compared to our findings are based on an early version of the reduced-complexity

Earth system model OSCAR which was constrained to the land sink without LULCC perturbation as estimated by DGVMs (Gasser and Ciais, 2013; Gasser et al., 2017). Later revised OSCAR versions, constrained to the net land flux as residual from fossil emissions, atmospheric growth, and the ocean sink, yielded higher LASC estimates (more similar to our study; Gasser et al. 2020). Note, the LASC of 0.8 PgC yr$^{-1}$ (0.84 PgC yr$^{-1}$) presented here is an estimate based on the TRENDY v8 model ouput combined with newer (TRENDY v9) output from SDGVM model (erroneous code in earlier versions caused a C loss

over the period ~1900-1970 mainly in semi-arid regions), while consistently using TRENDY v8 model output even results in a higher LASC of 0.9 PgC yr$^{-1}$ (0.85 PgC yr$^{-1}$). $f_{LULCC\_pd}$ is the DGVM-based $f_{LULCC}$ estimate that is most similar to bookkeeping results as expected (Sect. 1).

**Table 3.** Overview of global annual $f_{LULCC}$ estimates from this study, the ensemble of all 15 DGVMs and of two bookkeeping models (BLUE and H&N2017) from the annual global carbon budget (GCB2019; Friedlingstein et al. 2019), plus another recent bookkeeping estimate (Gasser et al., 2020). Emissions from peat fire and drainage were removed from the bookkeeping estimates to be better comparable to the DGVMs. Note that the error estimate of GCB2019's bookkeeping estimate of 0.7 PgC yr$^{-1}$ is an expert judgement, not direct model output. Minimum, maximum and mean with standard deviation refer to the model ensemble.

| Source | annual $f_{LULCC}$ (PgC yr$^{-1}$) | | | | | |
| --- | --- | --- | --- | --- | --- | --- |
| | 2018 | | | 2009–2018 | | |
| | Min | Mean ± 1SD | Max | Min | Mean ± 1SD | Max |
| $f_{LULCC\_trans}$ | 1.5 | 2.4 ± 0.6 | 3.4 | 0.8 | 2.0 ± 0.6 | 3.4 |
| $f_{LULCC\_pi}$ | 0.9 | 1.5 ± 0.5 | 2.4 | 0.5 | 1.2 ± 0.4 | 2.4 |
| $f_{LULCC\_pd}$ | 1.2 | 2.0 ± 0.8 | 3.5 | 0.7 | 1.6 ± 0.7 | 3.5 |
| GCB2019 – DGVMs | – | 2.3 ± 0.6 | – | – | 2.0 ± 0.5 | – |
| GCB2019 – bookk. models | 0.7 | 1.5 ± 0.7 | 2.1 | 1.0 | 1.5 ± 0.7 | 1.8 |
| Gasser et al. 2020 | – | 1.4 ± 0.4 | – | – | 1.4 ± 0.4 | – |

A closer look at the historical evolution of the three global $f_{LULCC}$ estimates reveals similarities, despite the substantial differences in their annual and cumulative quantities shown before. In particular, trends remain similar over time, with an in-

crease since the start of the simulations peaking in the 1950s and in the end of the simulation period (see multi-model means in Fig. 2a). Congruent patterns of $f_{LULCC\_pd}$ and bookkeeping mean values highlight the validity of our approach to investigate regions that are most sensitive towards choice of transient DGVM- vs bookkeeping-based estimates.

Throughout the 19th century, no differences are found between $f_{LULCC\_trans}$ and $f_{LULCC\_pi}$ (i.e. LASC around zero, Fig. 2b) indicating a negligible impact from environmental changes (i.e. $CO_2$ concentrations and climate). In line with this, the constantly



**Table 4.** Overview of global annual LASC estimates from this study, Friedlingstein et al. 2019 (GCB2019) and Gasser et al. 2020. LASC estimates from GCB2019 and Gasser et al. 2020 are based on two different versions of OSCAR, which is constrained by DGVM estimates. Minimum, maximum and mean with standard deviation refer to the model ensemble.

| | annual LASC (PgC yr$^{-1}$) | | | | | | | | |
| | 2018 | | | 2005-2014 | | | 2009–2018 | | |
| Published in | Min | Mean ± 1SD | Max | Min | Mean ± 1SD | Max | Min | Mean ± 1SD | Max |
|---|---|---|---|---|---|---|---|---|---|
| This study | 0.5 | 0.9 ± 0.3 | 1.4 | 0.1 | 0.7 ± 0.3 | 1.4 | 0.2 | 0.8 ± 0.3 | 1.4 |
| GCB2019 | – | – | – | – | 0.4 ± 0.3 | – | – | – | – |
| Gasser et al. 2020 | – | 0.8 ± 0.6 | – | – | – | – | – | 0.7 ± 0.6 | – |

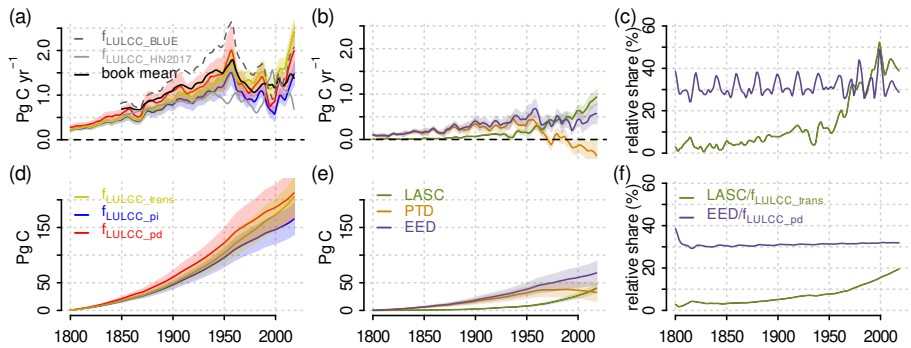

**Figure 2.** Multi-model means of smoothed global annual values (upper row) and cumulative sums (lower row) of $f_{LULCC}$ estimates, the Loss of Additional Sink Capacity (LASC), the 'Present-day' vs 'Transient' environmental conditions Difference (PTD), the Environmental Equilibrium Difference (EED), and the relative contributions of LASC and EED to $f_{LULCC\_trans}$ and $f_{LULCC\_pd}$ respectively from 1800 to 2018. Additionally, $f_{LULCC}$ from the bookkeeping models BLUE and H&N2017 as well as their average is plotted (data for GCB2019; not shown for cumulative sums due to shorter data coverage). For absolute values from this study also the 95% confidence intervals are shown. See Figs. 3 and 4 for individual model's results for $f_{LULCC}$ estimates and their differences, respectively.

higher and faster increasing annual and cumulative $f_{LULCC\_pd}$ (concomitantly PTD and EED, Fig. 2b,e) can be explained by higher C stocks due to their equilibration to present-day conditions rather than pre-industrial ones (compare Fig. 9 for historical C stock changes in the transient simulation). Similarly, the higher bookkeeping mean values compared to $f_{LULCC\_trans}$ and $f_{LULCC\_pi}$ up to the 1950s are attributable to their use of recent inventory-based C densities (Fig. 2a).

By the end of 19th century, annual and cumulative $f_{LULCC\_trans}$ estimates start to exceed $f_{LULCC\_pi}$ estimates. This can be related 270    to higher C stocks due to an accelerated atmospheric $CO_2$ increase where LULCCs leading to net loss in C stocks occurred (e.g. deforestation). Additionally, the aforementioned nature of the LASC as synergistic effect of changes in environmental conditions and any LULCC that occurred since the simulation start comes to play. As overall beneficial environmental alterations for C sequestration increased the potential C stocks (Fig. 9), the LASC steadily increased (Fig. 2b,e), reaching about





∼40% in recent annual and ∼20% in cumulative contributions to $f_{\mathrm{LULCC\_trans}}$ (Fig. 2c,f). Despite this LASC increase, global

annual and cumulative $f_{\mathrm{LULCC\_pd}}$ estimates still increase faster than the other estimates in the first half of the 20th century (EED and PTD remain increasing), indicating that synergistic effects of LULCCs with higher C stocks under present-day conditions still outweigh the amount of additional emissions accumulated by the LASC.

In the 1950s, global peaks in annual $f_{\mathrm{LULCC\_pi}}$ and $f_{\mathrm{LULCC\_pd}}$ estimates were observed. As these estimates neglect transient environmental conditions and do not include the LASC, this peaks simply relate to a strongly increased amount of LULCCs

depleting C stocks, in particular on C-dense land where historic environmental changes would have highly increased the potential C stocks (compare Fig. 9). The latter is highlighted by the simultaneous peak in EED which basically is the intersection of LULCCs with the difference in standing biomass and actual soil C stocks due to altered environmental conditions over the historic period (under pre-industrial versus present-day environmental conditions) and is independent from timing of LULCC occurrence.

The LASC becomes particularly evident after the 1950s, when the peak of converted C stocks by LULCCs was passed and a reduced amount of LULCCs decreasing C stocks caused strongly decreased annual $f_{\mathrm{LULCC\_pi}}$ and $f_{\mathrm{LULCC\_pd}}$ (and EED) estimates. By contrast, $f_{\mathrm{LULCC\_trans}}$ decreased only slightly, as the LASC grows largely due to a combination of large areas that have been transformed from natural vegetation to fast-turnover agricultural areas (not least during the 1950s peak in global LULCCs) and $CO_2$ levels accelerating their increase (Fig. A1). This accelerating increase of the LASC causes annual $f_{\mathrm{LULCC\_trans}}$ estimates

to surpass those of $f_{\mathrm{LULCC\_pd}}$ starting, for the multi-model mean, around 1960. PTD, as a consequence, becomes small, then negative (a small temporal lag is caused by the reduced subset of models used for PTD derivation). Around the same time, the LASC becomes larger than the EED, indicating that the foregone sinks by LULCCs outweigh the flux changes upon LULCCs under present-day vs pre-industrial environmental conditions. These changing differences in $f_{\mathrm{LULCC}}$ estimates over time highlight how sensitive the choice of $f_{\mathrm{LULCC}}$ definition is to considered timescales even on the global scale.


## 3.2    Differences in $f_{\mathrm{LULCC}}$ estimates on regional level

Where does the LASC occur, and which regions are most sensitive towards the investigated DGVM-based $f_{\mathrm{LULCC}}$ definitions (under constant pre-industrial and present-day or transient environmental conditions)? Compared to smoothed global curves, where signals average out, it must be expected that synergistic effects of C stock alterations in combination with the occurrence

and timing of LULCCs cause higher differences between the three $f_{\mathrm{LULCC}}$ estimations on regional scale. We assess these differences on a spatio-temporally explicit level using the RECCAP2 regions (Fig. A2) and show regional annual values of LASC, PTD and EED in Figures 5 to 7 (with corresponding cumulative estimates in Figs. A7 to A9; for a map refer to Fig. 11) and the underlying annual $f_{\mathrm{LULCC\_trans}}$, $f_{\mathrm{LULCC\_pi}}$ and $f_{\mathrm{LULCC\_pd}}$ in the appendix (Figs. A4 to A6; for a map refer to Fig. 11).

The largest sensitivity of cumulative $f_{\mathrm{LULCC}}$ towards choice of pre-industrial vs present-day environmental forcing is found in

vast stretches of the eastern USA, Southern Brazil, Eastern Europe to Central Asia, tropical Africa, India, China, and Southeast Asia (Figs. 7 and 11e). They reflect the areas of highest $f_{\mathrm{LULCC}}$ (Fig. 10a,c,e; compare increasing deviation of linear model





**Table 5.** Overview of global cumulative $f_{LULCC}$ and LASC estimates from this study, the ensemble of 15 DGVMs and of two bookkeeping models (BLUE (Hansis et al., 2015) and Houghton and Nassikas (2017)) from the annual global carbon budget (GCB2019, Friedlingstein et al. 2019), plus another recent bookkeeping estimate (Gasser et al., 2020). Emissions from peat fire and drainage were removed from the bookkeeping estimates to be better comparable to the DGVMs. Note that mean cumulative GCB2019 estimates are based on bookkeeping models, while their uncertainty is derived from DGVMs. LASC estimates from GCB2019 and Gasser et al. (2020) are based on two different versions of OSCAR, which is constrained by DGVM estimates. Minimum, maximum and mean with standard deviation refer to the model ensemble.

| | cumulative $f_{LULCC}$ (PgC) | | | | | |
| | 1750–2018 | | | 1850–2018 | | |
| Published in | Min | Mean ± 1SD | Max | Min | Mean ± 1SD | Max |
|---|---|---|---|---|---|---|
| This study, $f_{LULCC\_trans}$ | 118 | 215 ± 63 | 336 | 106 | 189 ± 56 | 290 |
| This study, $f_{LULCC\_pi}$ | 83 | 175 ± 55 | 287 | 72 | 149 ± 47 | 242 |
| This study, $f_{LULCC\_pd}$ | 147 | 224 ± 73 | 336 | 127 | 192 ± 64 | 292 |
| GCB2019 | – | 235 ± 75 | – | – | 205 ± 60 | – |
| Gasser et al. 2020 | – | 206 ± 57 | – | – | 178 ± 50 | – |
| | cumulative LASC (PgC) | | | | | |
| This study | 11 | 40 ± 15 | 65 | 11 | 40 ± 15 | 64.0 |
| GCB2019 | – | – | – | – | 20 ± 15 | – |
| Gasser et al. 2020 | – | 32 ± 23 | – | – | 31 ± 22 | – |

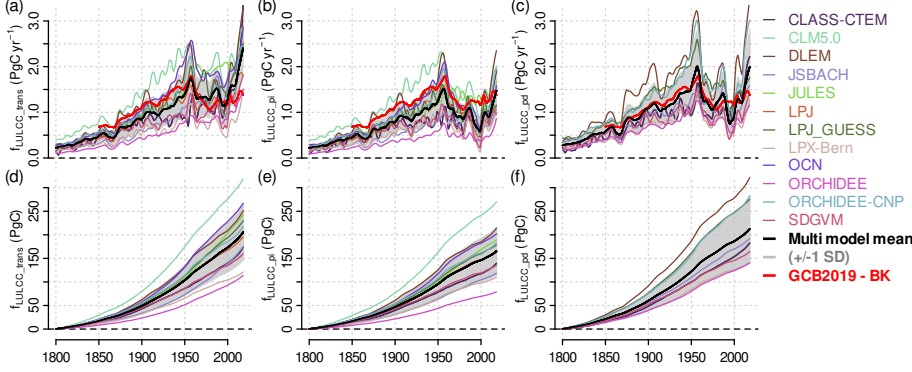

**Figure 3.** Smoothed global annual means (upper row) and cumulative sums (lower row) of $f_{LULCC\_trans}$ (a&d), $f_{LULCC\_pi}$ (b&e), and $f_{LULCC\_pd}$ (c&f) for the investigated DGVMs from 1800 to 2018. For the derivation formulas refer to Eqs. 1, 2 and 3, and for discussion on individual models refer to Sect. A1. $f_{LULCC\_pd}$ was not derived for CLM5.0, JULES, LPJ and OCN (compare Table 1). For comparison, we also included the GCB2019 bookkeeping mean (same values in all panels).





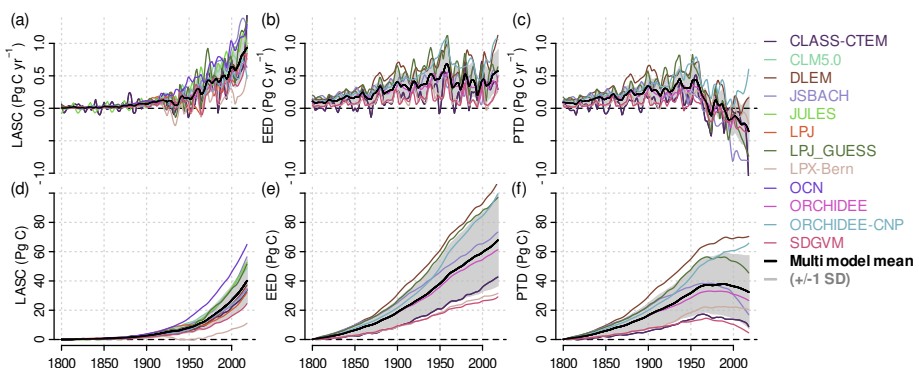

**Figure 4.** Smoothed global annual values (upper row) and their cumulative sums (lower row) of the differences in $f_{LULCC}$ estimates for the investigated DGVMs from 1800 to 2018: Loss of Additional Sink Capacity (LASC; panel a,d), Environmental Equilibrium Difference (EED; b,e) and 'Present-day' vs 'Transient' environmental conditions Difference in $f_{LULCC}$ (PTD; c, f). For the derivation formulas refer to Eqs. 4, 6 and 8, and for discussion on individual models refer to Sect. A1. EED and PTD were not derived for CLM5.0, JULES, LPJ and OCN (compare Table 1).

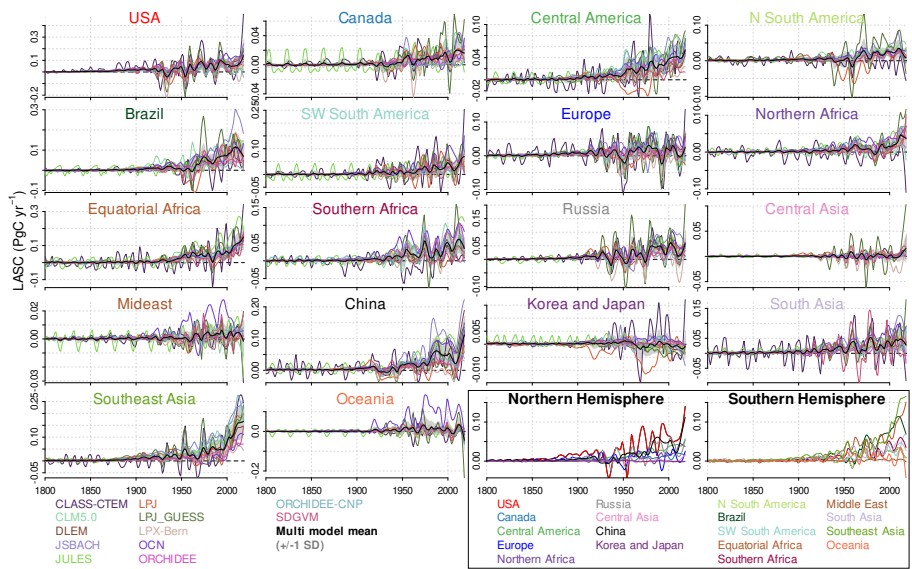

**Figure 5.** Regionwise smoothed annual Loss of Additional Sink Capacity (LASC) in the investigated DGVMs from 1800 to 2018, derived according to Eq. 4. For discussion on individual models refer to Sect. A1. The last two panels show regional ensemble means on uniform scale.

from 1:1 line with higher values), although there is some variation in the relative contribution of EED to $f_{LULCC\_pd}$ across regions that the global value of $\sim$35% (Fig. 2c,f) did not reveal (Fig. 8).

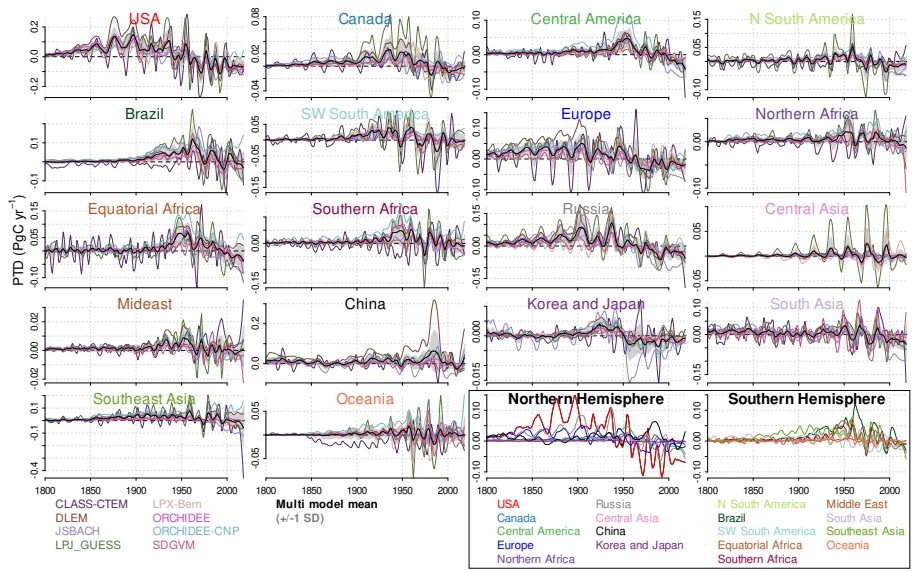

**Figure 6.** Regionwise smoothed annual 'Present-day' vs 'Transient' environmental conditions Difference in $f_{\text{LULCC}}$ (PTD) in the used models from 1800 to 2018, derived according to Eq. 6. For discussion on individual models refer to Sect. A1. PTD was not derived for CLM5.0, JULES, LPJ and OCN (compare Table 1). The last two panels show regional ensemble means on uniform scale.

The pattern of LULCC thus dominates the pattern of EED while ecosystem sensitivity to environmental conditions in general
seems to play a minor role. Particularly forested regions show positive changes in potential C stocks between 1800 and 2018
(Fig. 9) but not all sensitive regions show up in EED, e.g. remote rainforests have (so far) been less affected by clearing than
temperate forest regions. The very distinct region of negative cumulative EED in Central Europe (Fig. 11e) reflects relatively
increased $f_{\text{LULCC\_pd}}$ due to early and widespread reforestation (Fig. 10e). The associated C uptake with reforestation causes
globally wide-spread negative EED values in the last decade (Fig. 11f and 10f). Here we note that poor representation of
positive effects of recent large-scale reforestation programs on the C sink in China (Lu et al., 2018; Chen et al., 2019) in the
LUH2 data prevents EED (and also $f_{\text{LULCC}}$ estimates) to become negative in the affected regions. More strikingly, the last
decade saw the tropics to become more dominant in positive EED than other regions due to recent clearings. This shows, that
the choice of pre-industrial vs present-day environmental conditions can play a substantial role in regional $f_{\text{LULCC}}$ attribution:
EED cumulated >8 PgC in the USA, Brazil and Southeast Asia, >5 PgC in Russia, China, Equatorial Africa, Southern Africa,
and >2 PgC in Europe, Southwest South America and South Asia from 1800 until 2018 (Figs. 11e and A9).

### 3.2.1 Regions of positive loss of additional sink capacity - A lost carbon sink?

Not surprisingly, the regions of the largest LASC values are related to EED (compare Fig. 11a and e, and strong correlation
between LASC and EED in inlet Fig. 11e) and similar values to PTD (Fig. 11c) are in line with the cumulative LASC amounting
to about half of EED globally (Fig. 2e). But marked differences in patterns exist, which reflect that although the LASC is





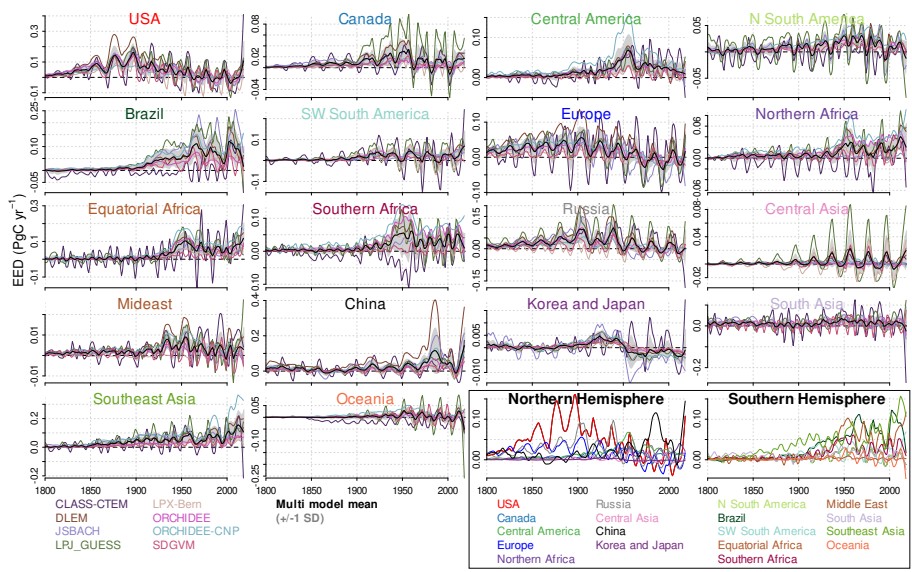

**Figure 7.** Regionwise smoothed annual difference between $f_{LULCC}$ under present-day and pre-industrial environmental conditions (Environmental Equilibrium Difference, EED) in the used models from 1800 to 2018, derived according to Eq. 8. For discussion on individual models refer to Sect. A1. EED was not derived for CLM5.0, JULES, LPJ and OCN (compare Table 1). The last two panels show regional ensemble means on uniform scale.

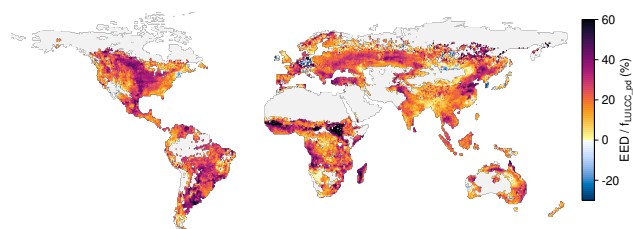

**Figure 8.** Multi-model means of the relative share of cumulative Environmental Equilibrium Difference (EED) to $f_{LULCC\_pd}$ from 1800 to 2018. Grid points with cumulative $f_{LULCC\_pd}$ <0.5 and >-0.5 were excluded from mapping.

driven by environmental differences, just as EED, it differs in causing fluxes on any area cleared in the past via the reference simulation seeing the potential vegetation within its pre-industrial extent. These differences are pronounced in the last decade (Fig. 11b,d,f): Regions, in particular forested ones, that were cleared between 1700 and the middle of the 20th century (when the accelerated $CO_2$ increase causes a strongly accumulating LASC) and stayed non-forested create emissions continuously during later times when the LASC is included and cause LASC to be larger than EED (i.e. negative PTD values) e.g. in

the eastern USA, Eastern Europe to Central Asia, and India. While EED is more relevant than the LASC for cumulative industrial-era emissions (stronger correlation in inlet Fig. 11e compared to Fig. 11f), the LASC heavily alters recent $f_{LULCC}$



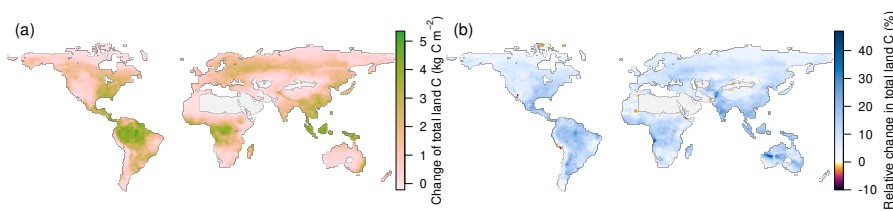

**Figure 9.** Multi-model means of absolute (a) and relative changes (b) in total carbon stocks (cTot; soil and vegetation carbon combined) from ∼1800 (average from 1800–1809) until today (average from 2009-2018) in the S2 simulation (including all environmental changes) within the vegetation extent of 1700. Grid points < 1 kgC m$^{-2}$ cTot in the later period were excluded.

estimates – Fig. 11b shows which regions would be attributed much higher emissions when the LASC is included in the $f_{LULCC}$

definition. Small areas exist where EED is larger than the LASC (i.e. positive PTD values) even for the recent decade: in the tropics (mainly Brazil, Tanzania, Indonesia), sub-tropics (Eastern China, Southern Australia), and in the transition zones from

temperate to boreal zone (Scandinavia, Russia). These regions experienced more recent LULCCs that reduced the C stocks, thus the LASC could only shortly accumulate. These regions would likely be attributed higher emissions by bookkeeping approaches (which are similar to $f_{LULCC\_pd}$) than by $f_{LULCC\_trans}$ from DGVMs. This highlights another difficulty especially in regional $f_{LULCC}$ attribution: as the LASC accumulates emissions caused by past LULCCs, recent LULCCs are given less weight in relative terms. This also applies to recent LULCCs reducing atmospheric $CO_2$ such as reforestation, which cannot

quickly compensate for past LULCC in approaches including the LASC, while they could in $f_{LULCC\_pi}$ and $f_{LULCC\_pd}$ estimates.

  Aggregated time series for the RECCAP2 regions reveal that the LASC started to increase ∼1850 in the USA, Russia and Southeast- and South Asia, ∼1900 in SW South- and Central- America and Southern Africa (Fig. 5). It then becomes even more pronounced ∼1950 in Brazil, Equatorial Africa and China, with the latter two and Southeast Asia showing a particular strong increase after 2000 (Figs. 5 and 11). Overall, the LASC accumulated to more than 4 PgC in the USA, Brazil, Equatorial

Africa and Southeast Asia, and to 2–4 PgC in China, Russia, SW South- and Central- America, Southern Africa and South Asia (Figs. 11 and A7). These high cumulative and annual LASC estimates mainly result from an initial high forest coverage and subsequent C losses in particular on areas where higher C stocks resulted from environmental changes over time (Sect. 3.4 and Fig. 9). Due to the different start of organized human agricultural, the forest clearings in the USA (mid of 19th century, though on forests with comparably low C stocks, Fig. A3) have caused an early LASC initiation, which cumulated to ∼5 PgC until

today (Fig. A7), while in Brazil, Equatorial Africa, China, and Southeast Asia, a much later onset of wide-spread LULCCs (beginning of 20th century) caused similar cumulative sums due to rapidly increasing and pronounced higher vegetation C stocks in the converted forests (strong response to $CO_2$ increase; Fig. A3).

### 3.2.2 Regions of negative loss of additional sink capacity - A gained carbon sink?

While it has been shown above that the LASC globally is a strong positive term adding almost 1 PgC yr$^{-1}$ to recent annual

$f_{LULCC}$, the LASC may be negative in some regions. Negative cumulative LASC estimates from 1800 onward are seen for wide





areas of Europe, small areas in Brazil (eastern parts) and Southern Africa (eastern parts), and, with lower quantities spread over Canada, Russia and China (Figs. A7 and 11a). Negative annual LASC estimates for the period 2009–2018 are observed in the same regions, but more wide-spread in Brazil and Southern Africa and striking negative values in the Ukraine (Figs. 5 and 11b). These negative LASC estimates can mainly be explained by LULCCs beneficial for C stocks (e.g. reforestation) on areas

that experienced beneficial environmental conditions afterwards, with a negative cumulative LASC indicating that the positive effects of LULCCs on the C stocks outweighed the effects of, mostly earlier, LULCCs that decreased C stocks. Note, this depends on the time LULCCs occurred, as the LASC accumulation periods differ, in their duration as well as the underlying transient environmental conditions. The strong negative cumulative and annual LASC estimates across France, Germany and Italy result from widespread reforestation after 1700, but also from the fact that the pre-industrial land use already had low

forest coverage due to pre-1700 deforestation (Klein Goldewijk et al., 2017), despite belonging to the forest biome. Most recent negative LASC values in the Ukraine can be linked to recultivation of post-Soviet abandoned agricultural land in particular in the Steppe zone (Smaliychuk et al., 2016). However, a negative LASC may also represent a negative climate change impact on C stocks (e.g. reduced precipitation) in areas where LULCCs decreasing C stocks happened (e.g. Iberian peninsula and eastern parts of South Africa).

The areas with a negative LASC are consequently attributed lower $f_{\mathrm{LULCC}}$ emissions to the atmosphere when the LASC is included in the calculation. If political reporting were based on DGVM-based $f_{\mathrm{LULCC\_trans}}$ estimates of the GCB, instead of a bookkeeping approach, these regions would 'profit' the most (be attributed less emissions). In other areas of widespread reforestation, most recent annual LASC estimates remain positive albeit decreasing, depending on how much the LASC has accumulated before as synergy between timing of LULCCs and later environmental C stock alterations. Here, a negative PTD

indicates that the LASC accumulated more than the difference of the actual fluxes upon detrimental LULCCs under transient vs present-day conditions (e.g. due to a long accumulation period), or that beneficial LULCCs caused smaller negative emissions in $f_{\mathrm{LULCC\_trans}}$ as compared to $f_{\mathrm{LULCC\_pd}}$.

### 3.3  Relative climate- and CO$_2$-induced $f_{\mathrm{LULCC\_trans}}$ components

As discussed (Sect. 2.2.2), patterns of CO$_2$ and climate changes may have very different effects on $f_{\mathrm{LULCC}}$ across the globe.

The mean simulated global vegetation C stock increased by ∼23% from 664 PgC to 815 PgC from 1800 until today, in both the S1 and S2 simulation (see Figs. 9 and A3 for maps and Fig. A1b for global estimates). The mean simulated global soil C stock increased from 1494 PgC to 1569 PgC (∼5%) in S1 and to 1553 PgC (∼4%) in the S2 simulation (see Fig. A1c). In line with the more pronounced soil C stock increase in the S1 simulation (excluding climatic changes), the general increase in cTot can mainly be attributed to an altered CO$_2$ exposure under rising atmospheric CO$_2$ (Lal, 2008). However, although climate

change (here roughly the last 100 years due to model assumptions) induces lower changes in C stocks on global scale, it has high impact on local and regional scale.

Climate change increased cTot mainly through vegetation changes in mid and high latitudes, which can be explained by increased temperatures leading to longer growing seasons, boreal expansion of biomes to mention a few (Peng et al., 2014; Piao et al., 2019) and increased precipitation in some regions (e.g. CMIP5 precipitation changes of last century in Becker



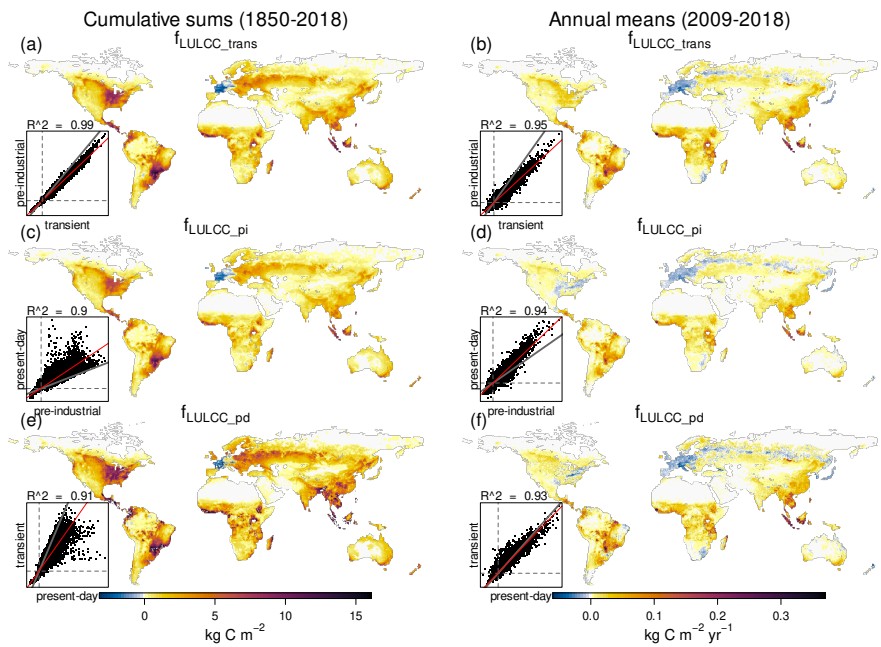

**Figure 10.** Cumulative sums from 1850 onward (left column) and annual means for 2009–2018 (right column) of $f_{LULCC\_trans}$ (upper row), $f_{LULCC\_pi}$ (middle), and $f_{LULCC\_pd}$ (lower row) averaged across the models. Additionaly, correlation plots between the pixel-wise estimations are shown; here the grey line represents the 1:1 line, the dashed grey lines depict zero lines, and the red line shows a fitted linear model. $f_{LULCC\_pd}$ was not derived for CLM5.0, JULES, LPJ and OCN models (compare Table 1).

et al. 2013; van den Besselaar et al. 2013). Negative climate change impacts on C stocks are mainly found across the tropics for vegetation and in most regions of the world for soil C. These negative climate-induced stock alterations likely relate to reduced precipitation amounts (e.g. Ren et al. 2013; van den Besselaar et al. 2013) with an increased frequency and intensity of droughts (e.g. Bastos et al. 2020), increased temperatures further increasing the vapor pressure deficit (potentially enhancing transpirational water losses) and increasing soil respiration and mineralization processes (reducing soil C stocks; Lal 2008;

Crowther et al. 2016; Davidson and Janssens 2006), and disturbances such as forest fires (Bowman et al., 2009; Archibald et al., 2018). The apparent dipoles in climate-induced vegetation and total C stock alterations in the USA and over Europe are most likely triggered by environmental changes during the 20th century with reduced stocks in USA and Southern Europe where precipitation decreased (and droughts happen more frequent) and higher stocks in the Eastern USA where precipitation widely increased (and droughts get less likely; e.g. Peterson et al. 2013; van den Besselaar et al. 2013) and northern Europe

due to global warming induced longer growing seasons (e.g. Keenan et al. 2014; O'Sullivan et al. 2020) .

In line with the homogeneously altered C stocks due to increased $CO_2$, spatial patterns of the $CO_2$-induced $f_{LULCC}$ component ($f_{LULCC\_CO_2}$) widely reflect $f_{LULCC\_trans}$, and thus LULCC activities, while the climate-induced $f_{LULCC}$ component is much more heterogeneously spread (Sect. 2.2.2 and Fig. 12). Highest $f_{LULCC\_CO_2}$ occurs in the tropics and in mid latitudes, where changes in vegetation C dominate the C pool changes and vast areas have been transformed by LULCCs that decreased



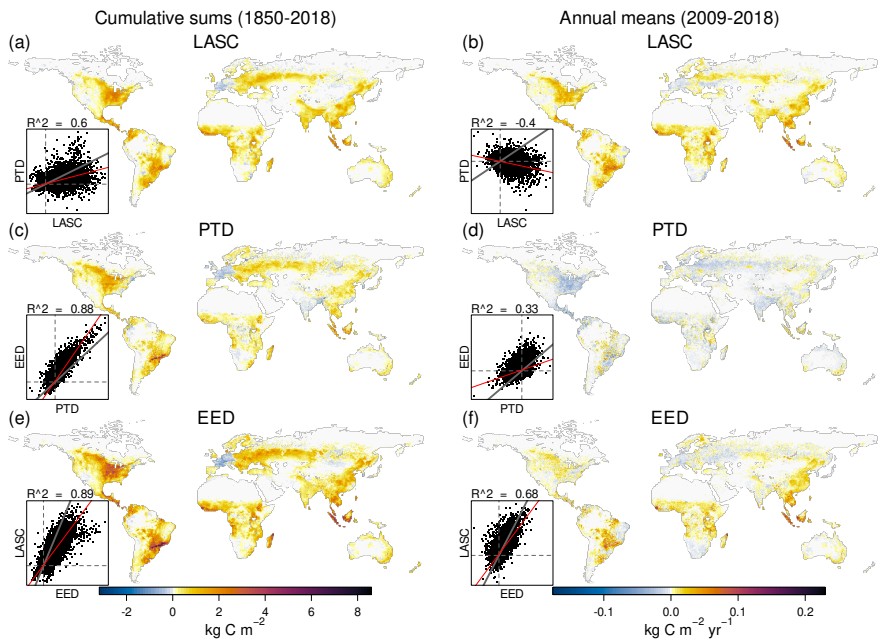

**Figure 11.** Cumulative sums from 1850 onward (left column) and annual means for 2009–2018 (right column) of the Loss of Additional Sink Capacity (LASC; upper row), the 'Present-day' vs 'Transient' environmental conditions Difference (PTD; middle row) and the Environmental Equilibrium Difference (EED; lower row) averaged across the models. Additionaly, correlation plots between the different pixel-wise estimations are shown; here the grey line represents the 1:1 line, the dashed grey lines depict zero lines, and the red line shows a fitted linear model. EED and PTD were not derived for CLM5.0, JULES, LPJ and OCN models (compare Table 1).

C stocks (Figs. A3 and 12a,b). Negative $f_{\text{LULCC\_CO}_2}$ estimates are mainly found where also $f_{\text{LULCC\_trans}}$ and can be explained by reforestation (for small areas in NE USA and NE Brazil, wide areas in Europe, parts of Russia, Georgia, Korea and Japan, and South Africa).

Although comparably low in absolute values, climate change induced alterations in $f_{\text{LULCC}}$ are much more heterogeneously spread over the globe and range from -23 to 28% with particular high alterations on areas with comparably low C stocks

(compare Figs. A3 and 12c,d,e,f). A reduced $f_{\text{LULCC\_Climate}}$ occurs where also vegetation C is reduced due to climate, mainly in the tropics and sub-tropics with particular hotspots in North East Brazil, the Mediterranean region, Southern and Eastern Africa, China, Southern Asia, Southwestern Australia, and Central America (the latter, despite higher vegetation C), and in the temperate zone, in Western USA and Mongolia. In contrast to this climate-induced $f_{\text{LULCC}}$ reductions, climate strongly increased $f_{\text{LULCC}}$ in particular in colder environments of higher latitudes and altitudes where higher C stocks resulted from

climate change (Sect. 2.2.2).

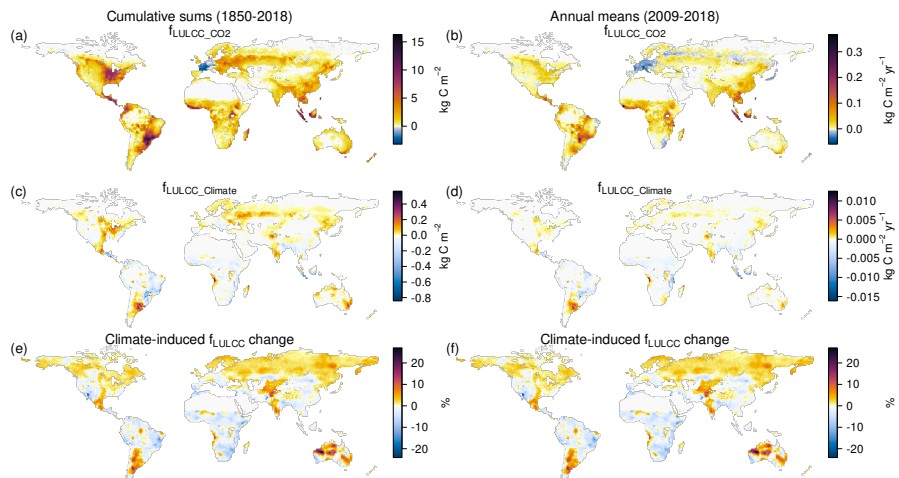

**Figure 12.** Cumulative sums from 1850-2018 (left column) and annual means for 2009–2018 (right column) of $f_{\text{LULCC\_CO}_2}$ (upper row), $f_{\text{LULCC\_Climate}}$ (middle) and percentage change in $f_{\text{LULCC\_trans}}$ due to climate change only (lower row; $100 \times (f_{\text{LULCC\_trans}} - f_{\text{LULCC\_CO}_2})/f_{\text{LULCC\_trans}}$). Grid boxes < 1 kgC m$^{-2}$ total C stock excluded from mapping.

## 4 Proposal for a standard $f_{\text{LULCC}}$ estimation

Previous chapters, for the first time, have shown that $f_{\text{LULCC}}$ patterns depend not only on the timing of occurrence and type of LULCCs, but also on the simulated time period and the assumptions on environmental conditions (with very diverse effects from climate alterations). Disregarding considerations from the natural land sink perspective, these results highlight the need for

a $f_{\text{LULCC}}$ estimate that is comparable over time and across space. For example, including the LASC in $f_{\text{LULCC}}$ estimates may be perceived as appropriate because LULCCs could have destroyed or created vegetation with long C turnover (e.g. deforestation or reforestation) leading to de- or increased C sinks (while current $f_{\text{LULCC}}$ reporting neglects such foregone sinks). However, including the LASC implies attributing fluxes to a region's emission budget that are partly a fate of history; in particular in the temperate regions, LULCCs detrimental to C stocks historically happened earlier compared to LULCCs increasing C stocks.

Thus, the committed emissions included in the LASC often have longer accumulation periods for detrimental as compared to beneficial LULCCs whose accumulation periods are more likely to be cut off at the simulation end (2018 in the GCB2019). The accumulation periods may further be altered if, over the historic period, various LULCCs occurred on the same area. This is further complicated because environmental changes over the historic period modified the LASC, with a widely accelerated accumulation rate in later periods due to higher, and faster increasing $CO_2$ concentrations but very heterogeneously spread

alterations by climatic changes. Thus, even for the same LULCC with the same accumulation duration, the LASC will be different dependent on timing and location of the LULCC.

To circumvent these issues, as could be desired in the political context, one could use $f_{\text{LULCC\_pi}}$ (which neglects transient conditions) as the base emissions and separately add an adapted LASC which is derived from defined reference accumulation





periods for different LULCC types. By using such reference periods, the LASC could fully be captured also for most recent LULCCs (may they act positive or negative on C stocks) and foregone sinks would be more equally counted. Additionally, to exclude LASC differences due to synergistic effects of environmental conditions and the timing of LULCCs, the adapted LASC accumulation periods should be independent of the actual time that LULCCs occurred and share the same reference conditions, for example the adapted LASC could always be modeled for the second half of the 21st century. Along these lines, it may be considered to calculate the adapted LASC based on $CO_2$-only simulations as here the impact of humans is more homogeneously distributed, while the spatially heterogeneous climate impact on $f_{\mathrm{LULCC}}$s, determined foremost by action outside the location of LULCCs, causes a questionable attribution of regional $f_{\mathrm{LULCC}}$ when compared across the globe (without even considering externalized $f_{\mathrm{LULCC}}$s e.g. due to remote market demand of food and timber; Lambin and Meyfroidt 2011; Meyfroidt et al. 2013). To detach $f_{\mathrm{LULCC}}$ estimates from the climate evolution, we argue to address the delineation of an adapted LASC in future studies. Such methodology could limit $f_{\mathrm{LULCC}}$ to locally determined factors (namely LULCCs) while still reflecting the foregone C sink capacity by human intervention.

## 5    Conclusions

Accurate quantification of the net carbon flux from land use and land cover changes ($f_{\mathrm{LULCC}}$) is essential, foremost to project carbon (C) cycle dynamics and estimate the strength of negative $CO_2$ emission technologies. However, $f_{\mathrm{LULCC}}$ can only be estimated by models – typically bookkeeping or dynamic global vegetation models (DGVMs) – and requires decisions on how to account for effects of environmental changes. We show that these decisions have major consequences for flux attribution, particularly at regional scale because C stocks evolve very heterogeneously in both space and time. DGVM estimates under present-day environmental forcing most closely resembled bookkeeping estimates (used in the annual global carbon budgets, GCBs) and are generally higher compared to $f_{\mathrm{LULCC}}$ under pre-industrial environmental conditions. This Environmental Equilibrium Difference (EED; accounting for ∼35% of global $f_{\mathrm{LULCC}}$ under present-day) is caused by higher C stocks, mainly in response to increased present-day atmospheric $CO_2$ and only to a smaller extent by climatic changes. Noteworthily, EED becomes negative in some regions, mainly due to environmental conditions decreasing C stocks (e.g. increased frequency and intensity of droughts and reduced precipitation). In the GCB, cumulative bookkeeping $f_{\mathrm{LULCC}}$ estimates are jointly published with DGVM-derived uncertainties under transient environmental conditions, which we show implies pronounced regional differences (named 'Present-day' vs 'Transient' environmental conditions Difference; PTD), strongly depending on the timing and placement of land use and land cover changes. We explain PTD values mainly by the loss of additional sink capacity (LASC), emissions due to destroyed C uptake potential that are only captured by the transient DGVM approach. In our multi-model mean for 2009–2018, a LASC of $0.8 \pm 0.3$ PgC yr$^{-1}$ accounts for ∼40% of recent global $f_{\mathrm{LULCC}}$ estimates of $2.0 \pm 0.6$ PgC yr$^{-1}$ (under transient conditions). The LASC causes strongly increased transient $f_{\mathrm{LULCC}}$ (>0.1 PgC yr$^{-1}$) where LULCCs detrimental to C stocks, such as deforestation, happened early within the simulated period (long accumulation period for lost potential C uptake; foremost in the USA) or later on areas with strong positive C stock response to environmental changes (e.g. in Brazil, Southeast Asia and Equatorial Africa). In contrast is transient $f_{\mathrm{LULCC}}$ strongly decreased where early reforestation occurred





on areas profiting from climate change (e.g. wide-spread in Europe). If environmental effects on potential C stocks should be accounted for fully, we argue to include the LASC into regional budgets, thereby highlighting the need for DGVMs. However, as LASC values derived by the common approach are widely independent of locally determined environmental changes but

depend on the arbitrary length of their accumulation period (defined by the simulated period, i.e. the start and end year of the simulations), it could be considered to derive an adapted LASC based on a defined reference period and homogeneously altered environmental conditions (such as only driven by $CO_2$ alterations).

*Code and data availability.* Scripts and data are available upon request from the corresponding author.

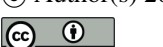



**Appendix A**

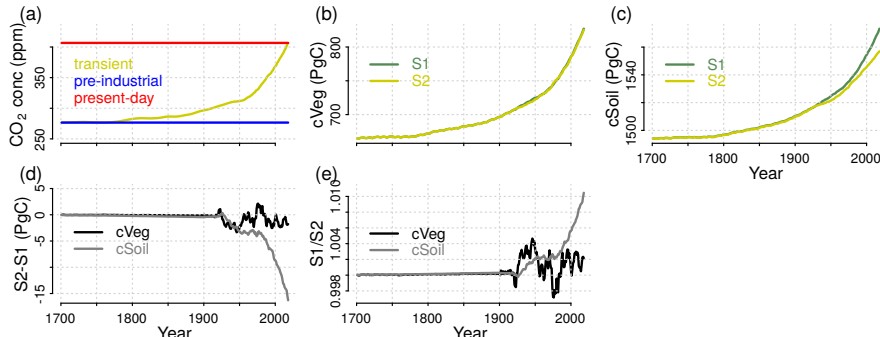

**Figure A1.** Global forcings of annual $CO_2$ fields and ensemble mean C stocks in vegetation and soil of the S1 (pre-industrial climate and transient $CO_2$) and S2 (transient climate and $CO_2$) simulation runs. Additionally, the differences and ratios in S1 and S2 C stocks in vegetation and soil are plotted.



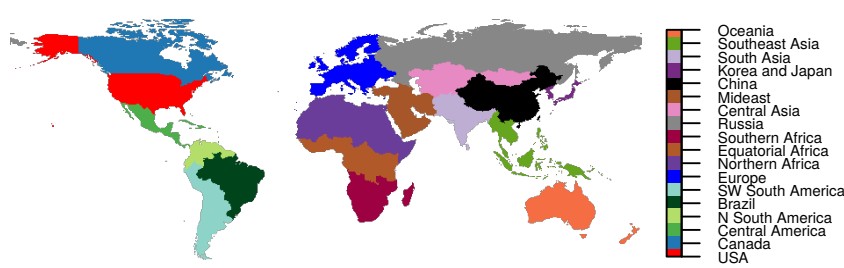

**Figure A2.** RECCAP2 global regions as defined in Tian et al. 2019.





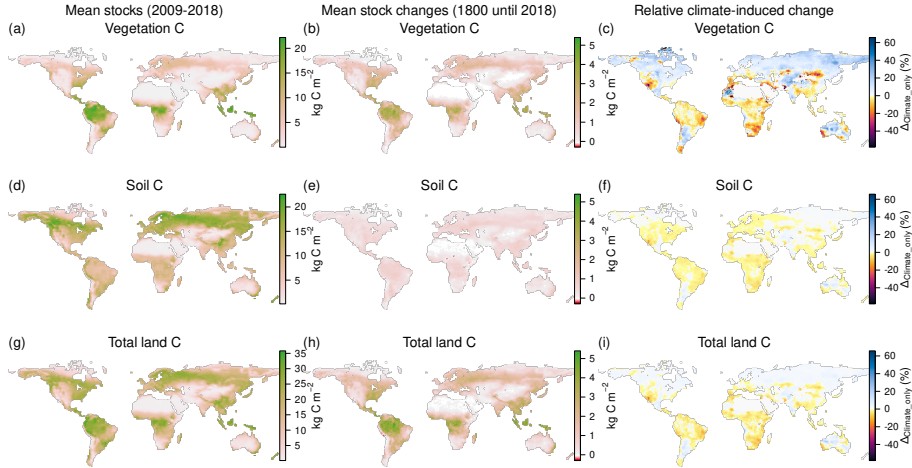

**Figure A3.** Ensemble mean C stocks from 2009–2018 in S2 simulation (left column; observed environmental conditions and pre-industrial land use and land cover), mean C stock changes between 1800 and 2018 (middle), and their climate-induced percentage changes (right column, $100 \times (S2 - S1)/S2$ of vegetation (upper row; for relative change, values $< -60\%$ were set to $-60\%$), soils (middle), and their totals (lower row). The relative climate-induced changes indicate additional (blueish) and reduced (reddish) stocks due to historic climate change (grid points $< 1$ kgC m$^{-2}$ total C stock excluded).





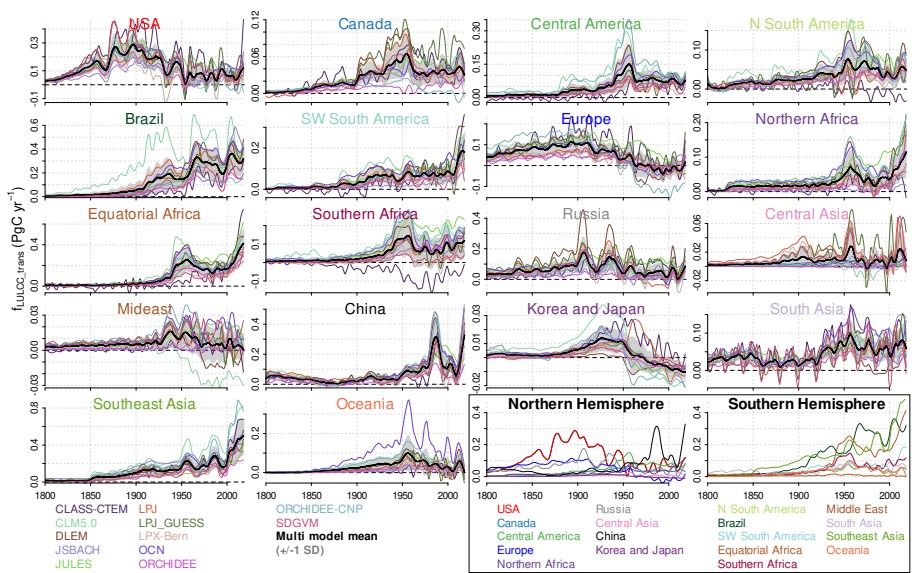

**Figure A4.** Regionwise smoothed annual $f_{\text{LULCC\_trans}}$ for different models from 1800 onward (compare Eq. 1). For discussion on individual models refer to Sect. A1. The last two panels show regional ensemble means on uniform scale.





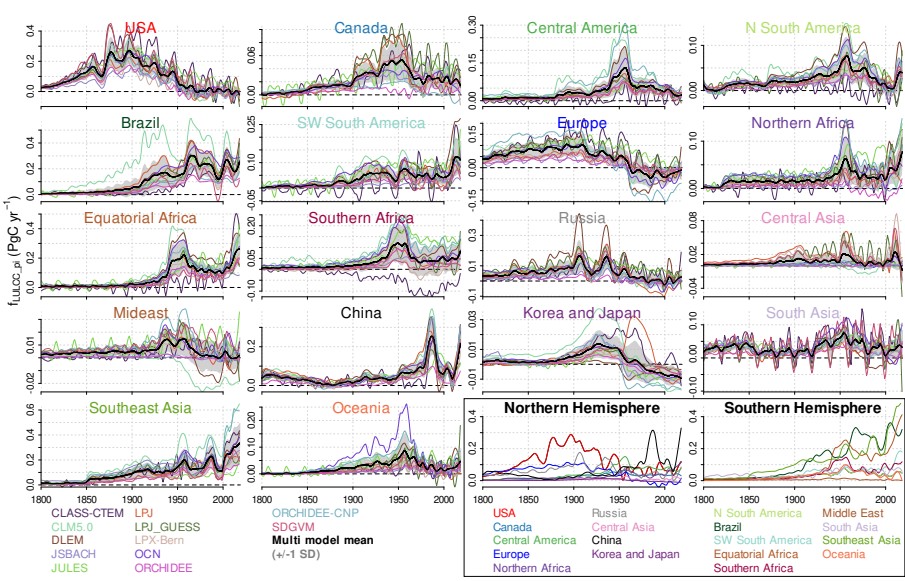

**Figure A5.** Regionwise smoothed annual $f_{\text{LULCC\_pi}}$ for different models from 1800 onward (compare Eq. 2). For discussion on individual models refer to Sect. A1. The last two panels show regional ensemble means on uniform scale.





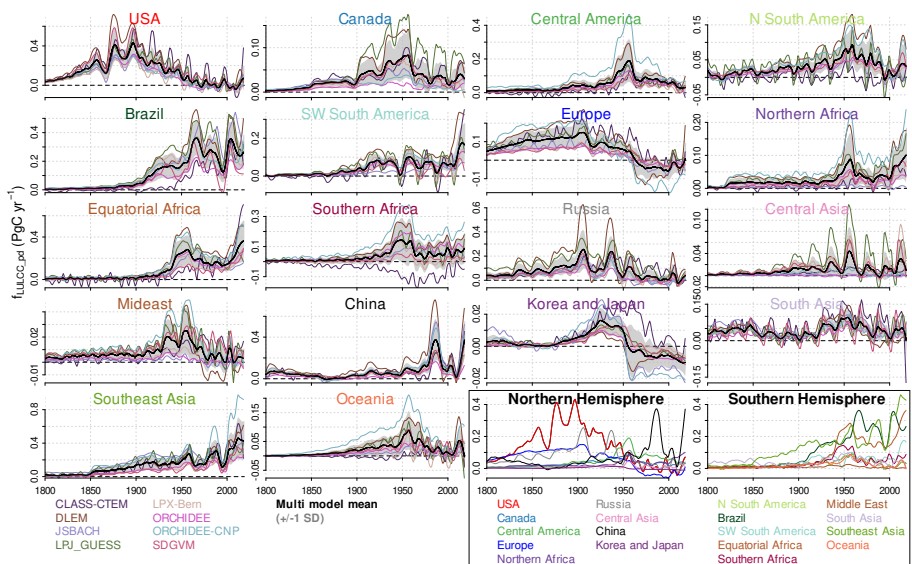

**Figure A6.** Regionwise smoothed annual $f_{\text{LULCC\_pd}}$ for different models from 1800 onward (compare Eq. 3). $f_{\text{LULCC\_pd}}$ was not derived for CLM5.0, JULES, LPJ and OCN models (compare Table 1). For discussion on individual models refer to Sect. A1. The last two panels show regional ensemble means on uniform scale.





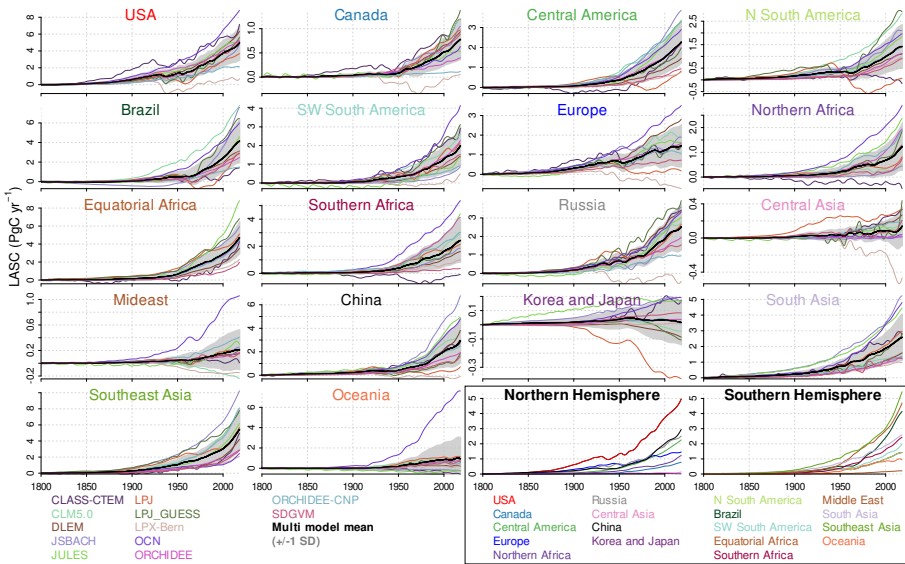

**Figure A7.** Regionwise smoothed cumulative Loss of Additional Sink Capacity (LASC) from 1800 onward (compare Eq. 4). For discussion on individual models refer to Sect. A1. The last two panels show regional ensemble means on uniform scale.





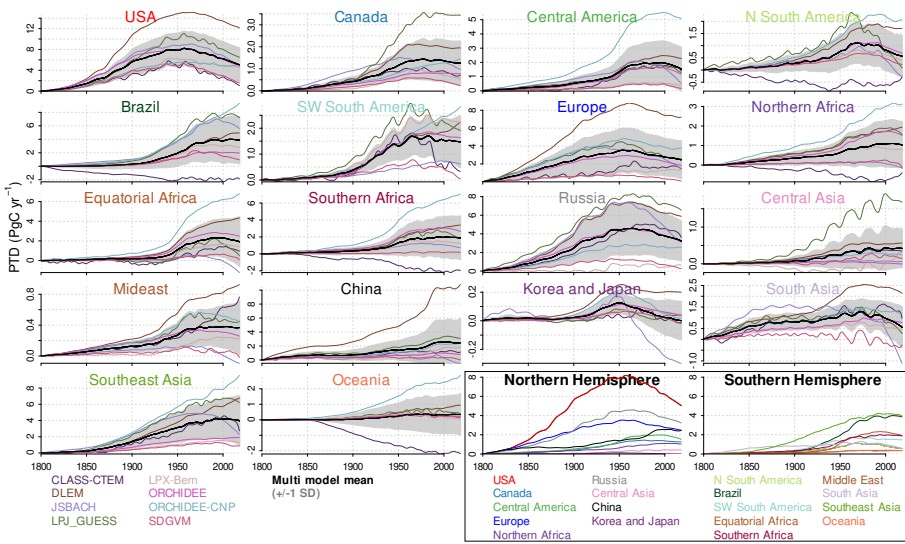

**Figure A8.** Regionwise smoothed cumulative 'Present-day' vs 'Transient' environmental conditions Difference in $f_{\text{LULCC}}$ (PTD) from 1800 onward (compare Eq. 6). PTD was not derived for CLM5.0, JULES, LPJ and OCN models (compare Table 1). For discussion on individual models refer to Sect. A1. The last two panels show regional ensemble means on uniform scale.



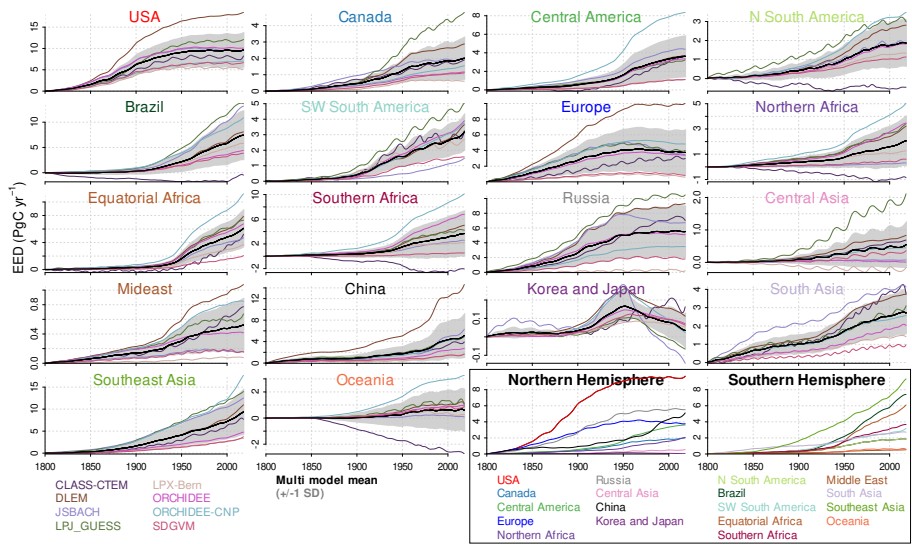

**Figure A9.** Regionwise smoothed cumulative difference between $f_{LULCC}$ under present-day and pre-industrial environmental conditions (Environmental Equilibrium Difference, EED) from 1800 onward (compare Eq. 8). EED was not derived for CLM5.0, JULES, LPJ and OCN models (compare Table 1). For discussion on individual models refer to Sect. A1. The last two panels show regional ensemble means on uniform scale.





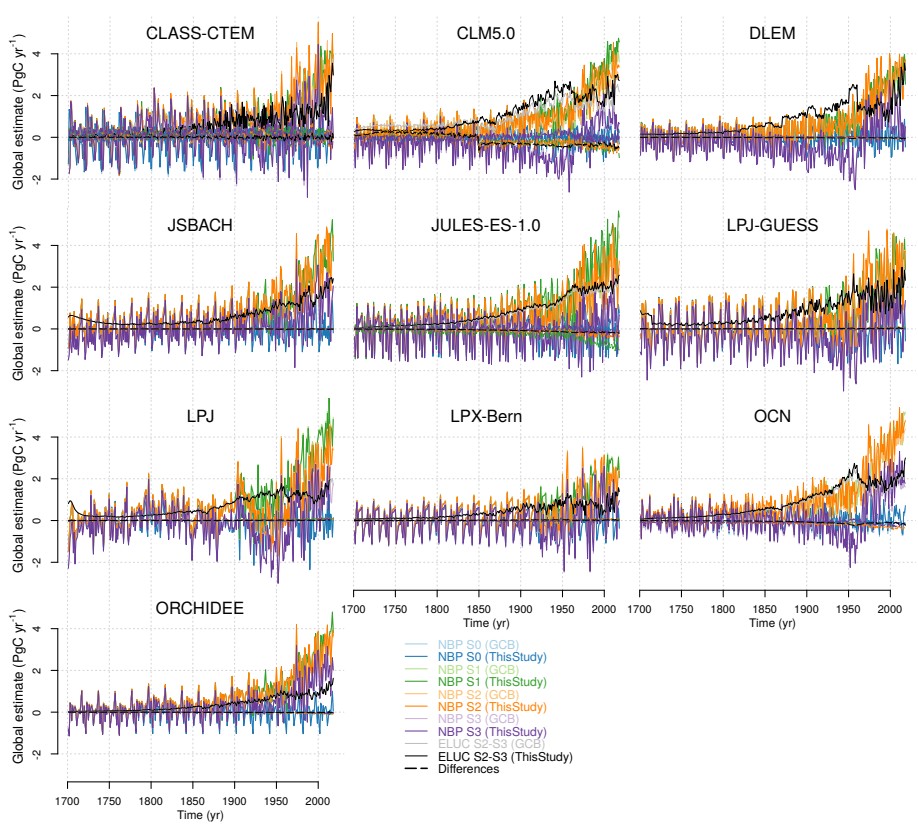

**Figure A10.** Comparison of global annual NBP of S0, S1, S2 and S3 simulation runs and derived $f_{\text{LULCC\_trans}}$ as aggregated in this study and published in the GCB2019 (Friedlingstein et al., 2019). The thick dashed lines (near or at zero) depict the differences (with respective colors). For the global values of this study, CDOs were used to convert NBP data per second to per year (multiplying seconds per day and, depending on the original temporal resolution, days per month or days per year), regrid data to the grid cell (multiplying with the area per grid cell). ORCHIDEE-CNP and SDGVM estimates were not shown since no data from the GCB2019 were available.

## A1    Model variability in $f_{\text{LULCC}}$ differences

The model spread in annual and cumulative $f_{\text{LULCC}}$ estimates and their differences (LASC, PTD and EED) has been shown to be large (compare Tables 3, 4 and 5), and increasing over time, in particular from 1950s onwards in Brazil, Northern Africa, Equatorial Africa and Southeast Asia (shaded areas in Figs. 3 and 4, showing the multi-model mean ±1 standard deviation). This can be explained by intertwining issues, such as the low quality of historical LULCC data (with different data bases), the simplified representation and uncertainty in the parameterization of of management and natural processes, uncertainties in soil and vegetation C stocks, and the lack of observtional constraints (Friedlingstein et al., 2019; Gasser et al., 2020; Lienert and Joos, 2018; Goll et al., 2015; Li et al., 2017). Additionally, a high interannual variability in the NBP data translated into a high variability of $f_{\text{LULCC}}$ estimates (Fig. A10) and their respective differences (even in the smoothed data; not shown). This e.g.





partly results some artificial periodic climate signal that might arise due to comparison of simulations with differently cycled
constant (present-day and pre-industrial) vs transient environmental conditions (e.g. on global scale, for relative share of EED
to $f_{\text{LULCC\_pd}}$ in Fig. 2c and, on regional scale, in Figs. 5 to 7 with pronounced oscillations in some regions).

Global EED and PTD were higher than in the other models for LPJ-GUESS, ORCHIDEE-CNP and DLEM and lower for
CLASS-CTEM, LPX-Bern and SDGVM. PTD and EED show highest model spread at the time of maximum LULCCs and
towards the end of the simulation period in particular in regions where vast areas of land were transformed (Brazil, Equatorial
Africa, Central, South and Southeast Asia).

A particular high model spread for global LASC at the end of the simulation period was found in Canada, N- and SW South
America, Brazil, Middle East, Korea and Japan, South and Southeast Asia and Oceania with particularly high estimates for
OCN, CLASS-CTEM, LPJ-GUESS and JSBACH models (Figs. 5 and A7).

High values in LPJ-GUESS likely result from high $f_{\text{LULCC}}$ estimates with pronounced inter-annual variability (particularly
prominent in Canada and Russia). This variability may be partially caused by stochastic components of the Globfirm fire
model, which was used in the TRENDY LPJ-GUESS runs, causing fire emissions not necessarily synchronous in time between
simulations runs.

High LASC estimates in JSBACH in Brazil and South and Southeast Asia can be explained by the strong positive response
of forest productivity to risig $CO_2$ concentrations in the model, and a consequently large LASC particularly upon clearing
of tropical evergreen forests. High EED and PTD estimates in ORCHIDEE-CNP in particular in Brazil, Southeast Asia and
Equatorial and Southern Africa, might result from accounting of phosphorus constraints on the biomass built-up under elevated
$CO_2$. ORCHIDEE-CNP simulates a more realistic sensitivity of plant productivity to elevated $CO_2$ than the version without
nutrients, ORCHIDEE (discussed in detail in Sun et al. 2020), but more models are needed to draw robust conclusions about
phosphorus effects on $f_{\text{LULCC}}$.

LPX-Bern showed very low LASC, EED and PTD estimates throughout the simulated period which result from low $f_{\text{LULCC}}$
estimates due to the exclusion of wood harvest and shifting cultivation, and, in particular in most recent decades, due to the
lack of tropical peatlands in the used configuration (for a detailed discussion refer to Lienert and Joos 2018).

Low EED and PTD estimates in CLASS-CTEM likely result from a model change that led to different S0 simulations (control)
for S1–S3 vs S4–S6 simulations which most probably also led to a pronounced variability and extreme values in some regional
estimates.

JULES showed a remarking high inter-annual variability for the LASC already in the early simulated period in particular in
Canada, SW South America, Middle East and Korea and Japan.

LPJ exhibited different IAV magnitudes for the pre-industrial and present-day land cover representation causing EED and
PTD unable to be calculated. The LPJ divergence in IAV may be due to differences in the carbon-climate sensitivity for
managed grasslands and croplands compared to natural ecosystems and further work is needed to understand the mechanisms
responsible.



*Author contributions.* JP and WO designed the study. JN and JP conducted preliminary analysis. AB, JN, JP, FH, KH, and TL contributed to processing and evaluation of the data. PA, AA, DG, JN, SL, DL, SL, PM, JM, BP, SS, MS, HT, APW, AJW, and SZ provided the TRENDY v8 data used in this study. WO processed the data, led the analysis and drafted the paper with contributions from all coauthors.

*Competing interests.* The authors declare that they have no conflict of interest.

*Acknowledgements.* We thank all people and institutions who provided the data used in this study and the 'Trends and drivers of the regional-scale sources and sinks of carbon dioxide' (TRENDY) modelling groups. AJW was supported by the Newton Fund through the Met Office Climate Science for Service Partnership Brazil (CSSP Brazil).





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
