# Peer review of "Modelled land use and land cover change emissions - A spatio-temporal comparison of different approaches"

_Earth System Dynamics, 2020_

## Author Comment (AC1)

**Response to Reviewer #1**

*In this study, the authors analyze three DGVM-derived fLULCC estimations for twelve models within 18 regions and quantify their differences as well as climate- and CO2-induced components. Results showed a global fLULCC of 2.0 ± 0.6 PgC per year for 2009–2018, of which ~40% are attributable to the LASC. Regional hotspots of high cumulative and annual LASC values are found in the USA, China, Brazil, Equatorial Africa and Southeast Asia, mainly due to deforestation for cropland. Distinct negative LASC estimates, in Europe (early reforestation) and from 2000 onward in the Ukraine (recultivation of post-Soviet abandoned agricultural land), indicate that fLULCC estimates in these regions are lower in transient DGVM- compared to bookkeeping-approaches. By unraveling spatio-temporal variability in three alternative DGVM-derived fLULCC estimates, our results call for a harmonized attribution of model-derived fLULCC. This study proposes an approach that bridges bookkeeping and DGVM approaches for fLULCC estimation by adopting a mean DGVM-ensemble LASC for a defined reference period. I would recommend this work for publication with few minor modifications.*

Thank you very much for the review and the recommendation for publication.

*Specific comment:*

*Line 130: More introduction about" gridded output" is needed. For example, the resolution of these data. Monthly data or Annual data?*

Thank you for this comment. We will add columns with information on the spatial and temporal resolution of output provided for each model to Table 1 and slightly modify the referencing text and table caption.

*Line 140: All abbreviations must be explained. For example, HYDE and FAO.*

Thank you, we will add the full names for HYDE and FAO at their first occurrence and slightly modify the respective sentence. Similarly, we will add the full name for RECCAP2 at first occurrence and to the caption of Figure A2. We could not find any other abbreviations that were not introduced.

*Line 154: 'the amount of precipitation in the Poyang Lake Basin' was not consistent with the caption.*

We are sorry, we can't relate this comment to our manuscript. Please specify.

*Line 166: Descriptions of three alternative fLULCC are not clear in the current version.*

Thank you, to ease the understanding of the different alternative $f_{LULCC}$ estimates, we will change the description in the methods section, assemble each equation within one line and add colored labels referring to the different $f_{LULCC}$ estimates into Figures 1, 3 and 5. Our suggestion for clarification:

NEW – 'We infer the three different DGVM-based $f_{LULCC}$s each from the differences in NBP of a simulation with and one without LULCCs (Eq. 1 to 3, see Table 1 for description of simulations S0 to

S6 and Fig. 1 for a schematic of resulting carbon fluxes). For example, we derive the $f_{LULCC}$ under transient environmental conditions by subtracting NBP in S3 from NBP in S2 (Eq. 1). Using yearly aggregated NBP values, $f_{LULCC}$ is derived for each DGVM, time step and grid cell under transient (subscript *trans*), constant pre-industrial (*pi*), and constant present-day (*pd*) environmental conditions from the TRENDY v8 simulations as follows:'

NEW – 'Here, a lower NBP in the simulation including LULCC (S3 to S5) compared to the one excluding LULCC (control, S0 to S2) represents a net flux of $CO_2$ out of the terrestrial biosphere into the atmosphere (emissions) due to LULCC causing a reduced C uptake or C losses.'

*Line 385: I have some serious concern about the assumption that the last 100 years due to climate change – clarify it?*

We are not sure we interpret the comment correctly (the sentence seems incomplete), but we will add a reference to Section 2.2.2 to explain better which climate (changes) our simulations capture. Due to the definition of the DGVM forcing within TRENDY v8, the climate of the first decades of the 20[th] century is recycled to infer earlier climatic conditions. This is a common procedure in model protocols since the trends in the physical climate before the 20[th] century are small (for information on TRENDY forcings for climate see CRU JRA data e.g. in Harris et al. 2014 and for $CO_2$ e.g. Joos and Spahni 2008). Because of this setup, the influence of climate change on NBP, and consequently C fluxes, as derived in this study depicts roughly the last hundred years, starting with the earliest decade of the 20[th] century. The validity of this approach is highlighted e.g. by proxy-based temperature reconstructions as published in Hegerl et al. (2019, *Environmental Research Letters*). To clarify, we will add the mentioned reference, modify the explanation given in Section 2.2.2 and suggest text changes as follows:

NEW - 'Note, within the TRENDY v8 simulations, pre-industrial and present-day climate forcing is defined as a recycling of climates in the earliest decades of the 20[th] and 21[st] century, respectively (see Sect. 1). Consequently, the climate change impact derived in this study roughly represents the last hundred years, which seems a reasonable approximation of the history, given that for example proxy-based temperature reconstructions cannot detect a warming earlier than the beginning of 20th century (Hegerl et al., 2019).'

NEW - 'The latter is highlighted by the simultaneous peak in EED which in essence is the intersection of LULCCs with the difference in standing biomass and actual soil C stocks due to altered environmental conditions over the last hundred years (under pre-industrial vs present-day environmental conditions; compare Fig. 2a with Fig. 2b and Sect. 2.2.2)'

*Eq 1,2,3: I really had difficulty in understanding these equations. I suggest the authors made them easy to follow in the revised manuscript.*

See answer to earlier comment. We will improve the description for the alternative $f_{LULCC}$ estimations by adding references to new Table 1 and Figure 1, additional labels and modified caption in Figure 1 (see next comment), assembling all equations within one line and by text modifications in the methods section (2.2.1).

*Figure 1 box 4 presents fLULCC differences, but no information about different line.*

We apologize as we forgot to name the assignment of line colors for the differences of the three $f_{LULCC}$ estimates in Figure 1. We will change the caption of Figure 1 as follows:

NEW - '[...] higher $f_{LULCC}$ (box 3: red line (present-day) higher than blue line (pre-industrial); yellow line (transient) increasing with time). [...] the Loss of Additional Sink Capacity (LASC; green line; Eq. 4), Environmental Equilibrium Difference (EED; purple line; Eq. 5) and `Present-day' vs `Transient' environmental conditions Difference (PTD; orange line; Eq. 6).'

####################################################################

In addition to changes resulting from the reviewer comments, we suggest the following changes:

- change to capitalized journal abbreviations in references and added doi that where missing

- add new Table 3 with overview of different estimates

####################################################################

New reference:

Hegerl, G. C., Brönnimann, S., Cowan, T., Friedman, A. R., Hawkins, E., Iles, C., Müller, W., Schurer, A., and Undorf, S.: Causes of climate change over the historical record, ENVIRON RES LETT, 14, 123 006, https://doi.org/10.1088/1748-9326/ab4557, 2019.

---

## Author Response (AR1)

**Response to Reviewer #1**

*In this study, the authors analyze three DGVM-derived fLULCC estimations for twelve models within 18 regions and quantify their differences as well as climate- and CO2-induced components. Results showed a global fLULCC of 2.0 ± 0.6 PgC per year for 2009–2018, of which ~40% are attributable to the LASC. Regional hotspots of high cumulative and annual LASC values are found in the USA, China, Brazil, Equatorial Africa and Southeast Asia, mainly due to deforestation for cropland. Distinct negative LASC estimates, in Europe (early reforestation) and from 2000 onward in the Ukraine (recultivation of post-Soviet abandoned agricultural land), indicate that fLULCC estimates in these regions are lower in transient DGVM- compared to bookkeeping-approaches. By unraveling spatio-temporal variability in three alternative DGVM-derived fLULCC estimates, our results call for a harmonized attribution of model-derived fLULCC. This study proposes an approach that bridges bookkeeping and DGVM approaches for fLULCC estimation by adopting a mean DGVM-ensemble LASC for a defined reference period. I would recommend this work for publication with few minor modifications.*

Thank you very much for the review and the recommendation for publication.

*Specific comment:*

*Line 130: More introduction about" gridded output" is needed. For example, the resolution of these data. Monthly data or Annual data?*

Thank you for this comment. We added columns with information on the spatial and temporal resolution of output provided for each model to Table 2 (earlier Table 1 due to rearranging to better match text flow) and slightly modified the referencing text (line 137) and table caption.

*Line 140: All abbreviations must be explained. For example, HYDE and FAO.*

Thank you, we added the full names for HYDE and FAO at their first occurrence and slightly modified the respective sentences (line 151 and lines 153-154). Similarly, we added the full name for RECCAP2 at first occurrence (line 207) and to the caption of Figure A2. We could not find any other abbreviations that were not introduced.

*Line 154: 'the amount of precipitation in the Poyang Lake Basin' was not consistent with the caption.*

We are sorry, we can't relate this comment to our manuscript. Please specify.

*Line 166: Descriptions of three alternative fLULCC are not clear in the current version.*

Thank you, to ease the understanding of the different alternative $f_{LULCC}$ estimates, we changed the description in the methods section (lines 162-175), assembled each equation within one line and added colored labels referring to the different $f_{LULCC}$ estimates into Figures 1, 3 and 5. Our clarification:

NEW (lines 162-167) – 'We infer the three different DGVM-based $f_{LULCC}$s each from the differences in NBP of a simulation with and one without LULCCs (Eq. 1 to 3, see Table 1 for description of simulations S0 to S6 and Fig. 1 for a schematic of resulting carbon fluxes). For example, we derive the $f_{LULCC}$ under transient environmental conditions by subtracting NBP in S3 from NBP in S2 (Eq. 1).

Using yearly aggregated NBP values, $f_{LULCC}$ is derived for each DGVM, time step and grid cell under transient (subscript *trans*), constant pre-industrial (*pi*), and constant present-day (*pd*) environmental conditions from the TRENDY v8 simulations as follows:'

NEW (lines 171-174) – 'Here, a lower NBP in the simulation including LULCC (S3 to S5) compared to the one excluding LULCC (control, S0 to S2 and S6) represents a net flux of $CO_2$ out of the terrestrial biosphere into the atmosphere (emissions) due to LULCC causing a reduced C uptake or C losses.'

*Line 385: I have some serious concern about the assumption that the last 100 years due to climate change – clarify it?*

We are not sure we interpret the comment correctly (the sentence seems incomplete), but we added a reference to Section 2.2.2 to explain better which climate (changes) our simulations capture. Due to the definition of the DGVM forcing within TRENDY v8, the climate of the first decades of the 20th century is recycled to infer earlier climatic conditions. This is a common procedure in model protocols since the trends in the physical climate before the 20th century are small (for information on TRENDY forcings for climate see CRU JRA data e.g. in Harris et al. 2014 and for $CO_2$ e.g. Joos and Spahni 2008). Because of this setup, the influence of climate change on NBP, and consequently C fluxes, as derived in this study depicts roughly the last hundred years, starting with the earliest decade of the 20th century. The validity of this approach is highlighted e.g. by proxy-based temperature reconstructions as published in Hegerl et al. (2019, *Environmental Research Letters*). To clarify, we added the mentioned reference (lines 256-257), modified the explanation given in Section 2.2.2 (lines 246-257) and made text changes as follows:

NEW (lines 253-257) - 'Note, within the TRENDY v8 simulations, pre-industrial and present-day climate forcing is defined as a recycling of climates in the earliest decades of the 20th and 21st century, respectively (see Sect. 1). Consequently, the climate change impact derived in this study roughly represents the last hundred years, which seems a reasonable approximation of the history, given that for example proxy-based temperature reconstructions cannot detect a warming earlier than the beginning of 20th century (Hegerl et al., 2019).'

NEW (lines 311-314) - 'The latter is highlighted by the simultaneous peak in EED which in essence is the intersection of LULCCs with the difference in standing biomass and actual soil C stocks due to altered environmental conditions over the last hundred years (under pre-industrial vs present-day environmental conditions; compare Fig. 2a with Fig. 2b and Sect. 2.2.2)'

*Eq 1,2,3: I really had difficulty in understanding these equations. I suggest the authors made them easy to follow in the revised manuscript.*

See answer to earlier comment. We improved the description for the alternative $f_{LULCC}$ estimations by adding references to new Table 1 and Figure 1 (lines 162-167), additional labels and modified caption in Figure 1 (see next comment), assembling all equations within one line and by text modifications in the methods section (2.2.1)(lines 162-167).

*Figure 1 box 4 presents fLULCC differences, but no information about different line.*

We apologize as we forgot to name the assignment of line colors for the differences of the three $f_{LULCC}$ estimates in Figure 1. We changed the caption of Figure 1 as follows:

NEW - '[...] higher $f_{LULCC}$ (box 3: red line (present-day) higher than blue line (pre-industrial); yellow line (transient) increasing with time). [...] the Loss of Additional Sink Capacity (LASC; green line; Eq. 4), Environmental Equilibrium Difference (EED; purple line; Eq. 5) and `Present-day' vs `Transient' environmental conditions Difference (PTD; orange line; Eq. 6).'

########################################################################

*Additional notes and references can be found at the end of the document.

**Response to Reviewer #2 (Wei Li)**

*The difference in Eluc estimates between DGVMs and bookkeeping models has been reported in many previous studies but never been quantified by decomposing into more specific fluxes like LASC. The authors filled this research gap using fractional simulations by various DGVMs. The analysis is very detailed and comprehensive with precise definitions of different LULCC fluxes and components and corresponding quantifications. It is a significant step in disentangling the Eluc components on top of Pongratz et al. (2014) and Gasser & Ciais (2013) and has important implications on the definition of Eluc in the global carbon budget and implementing climate mitigation measures. The manuscript is well written with clear description and detailed supplementary materials. I see no major flaw in this manuscript and thus suggest this work for publication with few minor revisions.*

Thank you very much for the review and the positive evaluation of our manuscript.

*I have some concerns that may need some clarification and discussion.*

*Although the LASC explains the Eluc difference between DGVMs and bookkeeping models, it would be better to address whether LASC exists in the real fluxes of carbon emission and sink, i.e. whether can be observed. If I understood correctly, positive LASC represents the loss of potential carbon sink and thus didn't physically exist, i.e. this part of CO2 wasn't released into the atmosphere. For the negative LASC like in reforestation, this part should be physically stored in the biomass or soil C pools, right?*

We agree with the reviewer and added corresponding discussions:

NEW (lines 71-78) - 'This example illustrates an interesting aspect of the LASC: It has been acknowledged that the LASC in its literal sense (a loss of carbon, positive LASC values) is an unrealized C uptake potential and is not reflected in any real change in atmospheric $CO_2$ concentration (Pongratz et al., 2014). However, as the LASC captures the foregone sinks a given LULCC event destroys and accumulates even in absence of further LULCCs, it manifests in the budget of atmospheric $CO_2$ as compared to a reference excluding LULCCs (Pongratz et al., 2014). In contrast to the theoretical nature of positive LASC values, negative values counted towards the LASC, for example due to reforestation, depict realized C uptake which is theoretically observable (though observations in the field are highly complex due to co-occurrence of natural carbon fluxes).'

NEW (lines 112-113) - 'On the other hand, as discussed, the lost sinks in DGVM-based $f_{LULCC}$ under realistic, transient environmental conditions do not correspond to observable fluxes.'

*As the authors stated, the timing of LULCC matters. Therefore, from my understanding, estimation of the accurate LASC for a specific LULCC event (a deforestation event in 1950 for an example) needs simulations similar to the S0 and S4 but using the climate and CO2 status when the LULCC event occurs (i.e. 1950) instead of the pre-industrial climate and CO2 status. Although the authors came to this point somewhere in the text, it would be better to emphasize this point explicitly.*

This is a very good idea which we incorporated into the revised manuscript. However, our idea is slightly different. Instead of recycling the climate and $CO_2$ status of the actual LULCC event for LASC

derivation, we argue to theoretically set all LULCC occurrences to the same time (e.g. 1950), thereby all accumulation periods would span the same period (and similar environmental changes) and LASC quantities would become independent of the actual timing of LULCCs. Moreover, to eliminate the variability resulting from the very heterogeneous effects of altered climatic conditions on NBP, we recommend to use $CO_2$-only forced simulations for LASC estimation (excluding climatic changes). Despite not fulfilling the modeler's need for a closed budget, this would make $f_{LULCC}$ attribution more independent of environmental changes and timing of LULCCs and lead to a more balanced quantification across space and time e.g. for regional or national $f_{LULCC}$ attribution. We changed several parts of the manuscript to emphasize the importance of the timing of LULCCs for $f_{LULCC}$ estimation as follows:

NEW (lines 6-10) - 'The LASC results from the impact of environmental changes on land carbon storage potential of managed land compared to potential vegetation, and accumulates over time, which is not captured in bookkeeping models. $f_{LULCC}$ from transient DGVM simulations, thus, strongly depends on the timing of land use and land cover changes mainly because LASC accumulation is cut off at the end of the simulated period.'

NEW (lines 478-490) - 'To circumvent these issues, as could be desired in the political context, one could derive $f_{LULCC}$ and the LASC based on simulations forced with the cycled climate and $CO_2$ conditions that occurred during the actual LULCC event. However, this would still result in differing accumulation periods and varying environmental conditions during and following a LULCC event. While the influence of the latter could be reduced using cycled pre-industrial or present-day environmental forcings, these neglect transient C stock changes. To consider the LASC but counteract spatial heterogeneity in $f_{LULCC}$ differences resulting from synergistic effects of environmental conditions and the timing of LULCCs, one could derive $f_{LULCC}$ and LASC from a defined reference period which is independent of the actual time that LULCCs occurred and shares the same reference conditions. For example, $f_{LULCC}$ and LASC could always be modelled for the second half of the 21st century, as here the environmental C stock changes have been amplified due to the accelerating increase of atmospheric $CO_2$ concentrations (alternative start times are of course conceivable). By using such reference period, the LASC could fully be captured also for most recent LULCCs (may they act positive or negative on C stocks) and foregone sinks would be more equally counted (same length of accumulation period with similar environmental changes)."

NEW (lines 522-527) - 'As LASC values derived by the approach so far taken in the GCB are widely independent of locally determined environmental changes (rather depend on globally determined climatic changes) and strongly dependent on accumulation periods (defined by the timing of LULCCs and the end year of the simulations), we argue for a $f_{LULCC}$ attribution that is more robust against choices of environmental drivers and accumulation period by using an adapted LASC, for example, based on a defined common reference period and homogeneously altered environmental conditions (such as only driven by $CO_2$ alterations).'

*The attribution to climate and CO2 in Sect. 2.2.2 is rather uncertain. Is there any observation data (e.g. FACE + warming experiment field data) that can be used to validate this attribution method?*

This is a good point. Experimental investigations have shown that there are interacting effects of $CO_2$ and e.g. temperature on biomass productivity (see e.g. Obermeier et al. 2017, *Nature Climate*

*Change*). However, within the TRENDY DGVMs, no significant interactions between these influencing factors on C stocks have been observed (e.g. Fernández-Martínez et al. 2019, *Nature Climate Change*). In line, a study using a fully-coupled DGVM found very low interaction between climate and $CO_2$, indicated by the almost equal effect size for the sum of individual effects ($CO_2$ fertilization, nitrogen deposition, climate warming and LULCC) compared to their combined effect (Devaraju et al. 2016, *Climate Dynamics*). Therefore, we assume that our approach is capable to properly approximate climate and $CO_2$ induced shares of $f_{LULCC}$. To highlight the underlying reasons, we carefully rephrased the description of the derivation of climate- vs $CO_2$-induced flux changes (also according to a comment of Reviewer #1) and add an explanation on the potential interaction (and included relevant references):

NEW (lines 237-242) - 'Here we note, interacting effects of elevated $CO_2$ concentrations and temperature or precipitation on biomass productivity (observed under experimental setups; e.g. Obermeier et al., 2017) might obscure this attribution (Lombardozzi et al., 2018). Nevertheless, the assessment of the relative contribution as done by this approach seams valid as no significant interactions between these influencing factors on C stocks were observed within the TRENDY ensemble (Fernández-Martínez et al., 2019) nor within a fully coupled single model investigation (Devaraju et al., 2016).'

*Fig. 5-7 show results from each DGVM, but not reported in the text. Could add one or two sentences to say which models always give high e.g. LASC in Fig.5 or all models are similar?*

Thank you. To focus on the main story, we shifted the region-wise plots showing individual models (Figs. 5-7) to the appendix and create new Figures 5 and 6 showing region-wise multi-model mean emissions estimates on unified y-scale (compare suggestions of Reviewer #3) for each hemisphere. In line, we discuss striking single model performances in the appendix section A1. Individual model performances of single models in the main manuscript are now only shown globally. According to your suggestion we have added the following sentence to the main manuscript:

NEW (line 280-283) - 'A widely congruent trend was also found across the DGVMs, while their absolute values partly differ strongly across models, for example global $f_{LULCC\_trans}$ and $f_{LULCC\_pi}$ from OCN is largely higher than in the other models, with estimates more than twice as large as the one from ORCHIDEE and LPX-Bern (Fig. 3, and Sect. A1 for a discussion on individual model results).'

*L29-30: "high-latitudes" usually refers to boreal region. I think much early agricultural expansion occurred in the temperate regions. May rephrase.*

Thank you. We apologize for this confusion. We totally agree that early agricultural expansion happened rather in the temperate than the boreal zone. The wording should actually have been 'higher latitudes' as compared to the tropics that are mentioned later in the sentence. To erase all potential confusion we changed the wording into 'mid-latitudes' (line 31).

*L248: bookkeping -> bookkeeping*

Thank you, we changed it (line 257).

*L253, L256: What are the numbers in the brackets?*

The numbers show the actual values with two decimal places. While there was only 0.01 PgC yr$^{-1}$ difference in LASC estimates ex- or including the erroneous model output from SDGVM (included e.g. in the GCB2019), the rounding to one decimal place caused differences of 0.1 PgC yr$^{-1}$. This gives us an absolute LASC difference greater than 10%, which we found noteworthy to be mentioned. We clarified this as follows:

NEW (lines 264-267) - 'Additionally, the inclusion of TRENDY v9 model output for SDGVM (for reasons refer to Sect. 2) caused, for example, a lower LASC of 0.8 PgC yr$^{-1}$ for 2009-2018 in this study (0.84 PgC yr$^{-1}$ with two decimal places) as compared to consistently using TRENDY v8 output, where the resulting LASC (usually rounded to one decimal place) is 0.9 PgC yr$^{-1}$ due to rounding of 0.85 PgC yr$^{-1}$.'

*L274: What are the pulses of the purple line in Fig. 2c?*

The pulses in the purple line result from (1) the internal climate variability, and (2) the combination of differently cycled forcings for the different simulations. Thanks to your question and a comment of Reviewer #3, we shifted parts of the discussion of Sect. A1 including an explanation for this to the main manuscript and improved the wording as well as referencing to figures:

NEW (lines 283-288) - 'Note, a high internal climate variability translated into a high interannual variability in NBP and consequently a high variability of $f_{LULCC}$ estimates (Figs. 3, 5 and A13) and of their respective differences (Figs. 4 and 6). For the differences in $f_{LULCC}$ estimates, some artefact might additionally arise due to comparison of simulations with different forcing cycles (e.g. on global scale, with periodic fluctuations in annual relative shares of EED to $f_{LULCC\_pd}$ in Fig. 2c and, on regional scale, in Figs. 6 and A4 to A6 with pronounced oscillations in some regions).'

*L312-313: Why reforestation increased fLULCC_pd?*

Thank you, we apologize for the misleading wording of the sentence. '[...] relatively increased $f_{LULCC\_pd}$' should refer to the higher emissions (may they be positive or negative) upon LULCCs under present-day as compared to pre-industrial environmental conditions. For clarification changed the wording as follows:

NEW (lines 346-347) - '[...] reflect increased C storage, and thus, a relatively stronger negative $f_{LULCC\_pd}$ due to early and widespread reforestation (Fig. 9e).'

###############################################################################

*Additional notes and references can be found at the end of the document.

**Response to Reviewer #3**

*The authors present a detailed analysis of multi-dynamic-ecosystem-model estimates of LULCC emissions under different initial and environmental conditions, with some comparison with recent bookkeeping approaches. A main focus is the contribution of the "Loss of Additional Sink Capacity" (LASC) to emissions, which can be best estimated by dynamic ecosystem models with transient environmental conditions. The authors conclude that LASC is considerable, and has some regional variation. Furthermore, they show that CO2 effects dominate the contributions to LASC over climate effects. They also raise the question of whether LASC is appropriate for a harmonized attribution of LULCC emissions.*

*Overall review*

*This is an important contribution to LULCC and carbon/climate science and presents interesting results from the trendy model ensemble. There are a few main issues that need to be addressed, however, prior to publication. These are summarized here, with further detail following:*

Thank you for this thorough review of our manuscript and the positive evaluation. We followed your suggestions which considerably improved the manuscript.

*1) There seems to be a better way to calculate the CO2 vs climate estimates. You have simulations that you can difference directly, rather then using the ratio method that has limitations.*

To our knowledge, the proposed ratio method is the only possible option to derive climate vs $CO_2$ related alterations in $f_{LULCC}$ based on available data (TRENDY v8). Because S0, S1, and S2 simulations have fixed pre-industrial LULCC forcing, their comparison does not allow to derive $f_{LULCC}$. Additionally, S1 is the only simulation where only $CO_2$ but not climate conditions are transient while no corresponding simulation exists that includes LULCC. Therefore, the available TRENDY simulations do not allow to directly derive climate vs $CO_2$ related alteration in $f_{LULCC}$. We now better explain this in the revised manuscript. For the details, please refer to the answers given below (in particular the answer to comment on lines 220-240).

*2) The results and discussion of regional LASC are confusing and not to the point. These can be better constrained and made more clear with main points in mind.*

See below and responses to the other reviewers how we improved the results and discussion. In particular, we changed the structure and wording in Sect. 3.1, 3.2, 3.2.1 and 3.2.2 (see response to your comment on lines 296-320 and the comment on lines 321-352).

*3) The suggested "harmonized attribution" of LULCC emissions is not consistent with the paper findings and doesn't differentiate between science and policy. The real issue here is full scientific estimates of emissions, as the authors clearly show require the inclusion of LASC, vs attribution of LULCC emissions for policy purposes, which is a political issue. This paper isn't about harmonizing different model approaches, but rather clarifying and understanding their differences. This provides scientific bases for accurate estimation of emissions, and provides info for political decisions regarding accounting for attribution policy, but doesn't show an answer to the "correct" accounting for policy or whether/how the different approaches should be "harmonized."*

Thank you, we agree that we did not differentiate enough between science and policy relevance of our findings. Instead of referring to a 'correct' or 'harmonised attribution', we now talk about an attribution that is more robust against choices of environmental drivers, historical timing of LULCCs and the accumulation period. With this in mind, we incorporated your suggestion on the reference period (and a suggestion given by Reviewer #2) to give various possibilities for a more balanced estimation of $f_{LULCC}$ and LASC (and highlight the need for future research, in particular, to investigate a suited reference period). Additionally, we now highlight the policy relevance where needed to enable a better differentiation for the reader. The revised manuscript, at several places, includes respective assignments (see responses below for more details, in particular answers given to your comment on lines 416-445 and comment on lines 467-472).

*Specific comments and suggestions*

*Abstract*

*The abstract focuses more on the argument for LASC and DGVMs, than on the results of the study. The argument is better suited for the discussion, with a brief statement in the abstract stating that you use DGVMs to estimate LASC. Then you can add more findings/explanation to the abstract, such as temporal transitions in PTD that support the DGVM vs bookkeeping difference statements.*

We carefully rephrased the abstract in several places to avoid the impression that we make recommendations for policy and/or science. However, the LASC is one of the most important differences between DGVM and bookkeeping estimates and has never been analyzed over time and space and many models, which justifies its prominence in the abstract; we clarified this as well through rephrasing and shorten the LASC description to also include the temporal transition in PTD around 1960 (though not namely, as the term is not introduced in the abstract):

NEW (lines 6-14) - 'The LASC results from the impact of environmental changes on land carbon storage potential of managed land compared to potential vegetation, and accumulates over time, which is not captured in bookkeeping models. $f_{LULCC}$ from transient DGVM simulations, thus, strongly depends on the timing of land use and land cover changes mainly because LASC accumulation is cut off at the end of the simulated period. To estimate the LASC, $f_{LULCC}$ from pre-industrial DGVM simulations, which is independent of changing environmental conditions, can be used. Additionally, DGVMs using constant present-day environmental forcing enable an approximation of bookkeeping estimates.'

NEW (lines 16-18) – 'Around 1960, the accumulating nature of the LASC causes global transient $f_{LULCC}$ estimates to exceed estimates under present-day conditions, despite generally increased carbon stocks in the latter.'

*line 8:*

*"arbitrary chosen" is unnecessary.*

We deleted 'arbitrary chosen', see comment above.

*lines 10-11:*

*Constant environmental forcing does not generate a condition independent of timing and legacy of lulcc. It generates a condition independent of changing environmental conditions, which is how this should be described.*

*Ensuring that the lulcc magnitude and trajectory are identical across simulations, along with constant environmental forcing, creates a condition almost independent of the timing and legacy of lulcc, as long as the transient dependence isn't the question. Full independence in this case requires that the biogeochemical dynamics are also identical across simulations, such as may be possible when a single model is used. But this is not possible across different models because each model will respond differently to the same exact lulcc (which is also very difficult to achieve across models), even if the environmental forcing is identical.*

*I think you mean this in the context of a single-model experiment, with the exact same lulcc and environmental forcing, and not focused on how the LULCC trajectory determines emissions (i.e., path dependence), but this needs to be qualified here as it sounds more general.*

Thank you for this interesting point. We agree that $f_{LULCC}$ is not fully independent of timing of LULCCs and legacy effects due to different biogeochemical dynamics implemented in the different DGVMs. We changed the wording according to your suggestion to avoid misunderstanding (see comment above and further comments below).

*lines 21-23:*

*I am not sure that your analysis calls for this, nor that your approach bridges the two. The issue is a full scientific accounting of emissions vs a partial accounting for policy purposes. While you have evidence for a full scientific accounting, you still have little basis for a partial-LASC approach.*

Thank you, we agree that our original statement was overstated. We changed the wording to better match the main outputs of this study, as follows:

NEW (lines 21-26) - 'Our study unravels the strong dependence of $f_{LULCC}$ estimates on the time a certain land use and land cover change event happened to occur, and on the chosen time period for the forcing of environmental conditions in the underlying simulations. We argue for an approach that provides an accounting of $f_{LULCC}$ that is more robust against these choices, for example by estimating a mean DGVM-ensemble $f_{LULCC}$ and LASC for a defined reference period and homogeneous environmental changes ($CO_2$-only).'

*Introduction*

*lines 28-30:*

*awkward sentence. delete "e.g." and use three clauses: deforestation...high lats, tropical deforestation, and recent forest expansion in high lats.*

Thank you, we changed the wording according to your suggestion and a comment of Reviewer #2 as follows:

NEW (lines 30-31) - '[...] in particular through deforestation driven by early agricultural expansion in mid-latitudes, recent tropical deforestation, and recent forest expansion in mid- and high-latitudes (Klein Goldewijk et al., 2011).'

*line 31:*

*"...contributed approximately one-third of global anthropogenic..."*

Changed accordingly (line 33) .

*line 35:*

*delete "in line." then: flulcc may also gain...*

Changed accordingly (line 37).

*line 42:*

*"...but also change in..."*

Changed accordingly (line 44).

*line 50:*

*delete "allow to"*

Changed accordingly (lines 51-52).

*lines 69-72:*

*The language here starts to confuse the definition and example of LASC. It sounds like just having the potential vegetation or the foregone sink is part of LASC, in addition to the difference of environmental effects on the managed vs unmanaged vegetation. These two sentences should be reworded to clarify that "assuming potential vegetation" and "capture the foregone sinks a given LULCC event destroys" do not obfuscate the definition of LASC.*

Thank you, we reworded this paragraph considering your comment and a comment of Reviewer #2 as follows:

NEW (lines 69-78) - 'Thus, even after the instantaneous emissions of the deforestation event may have ceased, deforestation continues to alter the $f_{LULCC}$ since the reference simulation assumes the potential vegetation cover in the absence of LULCCs, and simulates its response to environmental changes. This example illustrates an interesting aspect of the LASC: It has been acknowledged that the LASC in its literal sense (a loss of carbon, positive LASC values) is an unrealized C uptake potential and is not reflected in any real change in atmospheric $CO_2$ concentration (Pongratz et al., 2014). However, as the LASC captures the foregone sinks a given LULCC event destroys and likely accumulates even in absence of further LULCCs, it manifests in the budget of atmospheric $CO_2$ as compared to a reference excluding LULCCs (Pongratz et al., 2014). In contrast to the theoretical nature of positive LASC values, negative values counted towards the LASC, for example due to reforestation, depict realized C uptake which is theoretically observable (though observations in the field are highly complex due to co-occurrence of natural carbon fluxes).'

*lines 71-72:*

*this statement is not correct. there is no guarantee that the change in environmental conditions is increasing LASC at the time of change. so "same direction" does not make sense. I think you are still referring to the example here, but this is a more general concluding statement.*

*it is more correct to state that these emissions "can" accumulate in the absence of further LULCC, depending on the environmental conditions. which you then give examples of in the next paragraph.*

Thank you, we agree that the statement, referring to the example before, was not generally valid. We changed the wording according to your suggestion (see response to previous comment).

*lines 98-99:*

*I don't agree that constant conditions create a condition independent of lulcc timing. regardless of the forcing, the time since disturbance makes a difference, especially in the context of DGVMs where ecosystems are not static. A subsequent LULCC has a different emission depending on the period between it and the previous LULCC. And an event early in time on 'pristine' land may affect a different biomass then one later in time, even if the conditions are identical. Even with a spun-up model, the biomass may not remain constant during a simulation, and depending on the resolution and application of LULCC, a later LULCC may or may not be applied consistently to 'pristine' or 'managed' area (and corresponding biomass), or some combination of the two.*

*I think this description should be changed to "independent of environmental trends" or something like this.*

Thank you. We agree that, although based on fixed environmental forcings, $f_{LULCC}$ derived under present-day and/or pre-industrial conditions is not totally time-independent given the reasons you stated. We altered the description accordingly:

NEW (lines 104-105) - 'However, as $f_{LULCC}$ quantities derived under constant present-day conditions are independent of long-term environmental trends [...]'

*line 108:*

*delete "in line"*

Thank you, we deleted 'in line' (line 108) .

*lines 109-110:*

*unclear: "and consequently into the natural land c sink." how is this related to 'included or excluded?'*

We apologize for the unclear wording and modified the sentence to make it clearer:

NEW (lines 114-118) - 'It needs to be decided whether the LASC should be included or excluded (as argued e.g. in Gasser and Ciais 2013; Gasser et al. 2020) as part of $f_{LULCC}$ and consequently be counted towards the terrestrial C sink or not.'

*Data and methods*

*lines171-172 and 178-179 and 194-195:*

*These really seem like a single equation each.*

Thank you. We put all related equations into single lines (Eq. 4-6, 8).

*again, this isn't true. timing and legacy of LULCC are inherently critical and determining elements in any transient LULCC simulation. these simulations are indifferent to environmental trends or changing environmental conditions.*

Thank you, we improved the description according to your suggestion:

NEW (line 195-196) - 'The latter two are derived under constant environmental forcing, meaning that both are indifferent to long-term environmental trends [...]'

*"...C stock changes..."*

Changed accordingly (line 227).

*Why did you do it this way, particularly for climate effects? You can get co2+ndep-related emissions directly by subtracting S0 and S1, which would be interesting to compare with your estimate of co2 only emissions using this ratio method.*

*And you can get the climate-related emissions directly by subtracting S1 and S2, including interactions with CO2, without using this indirect ratio method. and your assumption regarding zero interactions between climate and co2 is not sound because S2 includes both at the same time, and S1 includes only co2. I suggest you use the actual nbp difference of these 2 sims to estimate the climate effects.*

Indeed, subtracting NBP from S0 and S1 simulations enables the isolation of effects related to $CO_2$ concentration and nitrogen deposition on the biospheric carbon fluxes. Unfortunately, as the land use/cover distribution is fixed to the pre-industrial state in both simulations, a derivation of the effects of $CO_2$ and nitrogen deposition effects on $f_{LULCC}$ is not possible from these simulations. We added a sentence to clarify this and slightly changed the respective paragraph:

NEW (lines 225-227) - 'These simulations do not include transient LULCCs and can therefore not directly be used to estimate the climate vs $CO_2$ related alteration of $f_{LULCC}$. However, [...]'

Similarly, subtracting NBP from S1 and S2 enables isolation of climate effects on biospheric C fluxes, while $f_{LULCC}$ alteration by climate can not directly be derived this way. In Figure A3 we show the relative change of C stocks in S1 vs S2 simulations for comparison.

It is true that the S2 simulation includes climate and $CO_2$ related effects on NBP at the same time. Thus, the statement 'Synergies [...] are assumed zero in this case.' was confusing. However, the statement is referring to our ratio approach, with the climate-induced $f_{LULCC}$ share being derived as the total $f_{LULCC}$ minus $f_{LULCC\_CO2}$ (the latter being the flux share if only $CO_2$ concentrations would have changed). In line with the response to a comment of Reviewer #2 we added an explanation on the underlying assumption. Additionally, to eliminate confusion we changed the wording as follows:

NEW (lines 237-242) - 'Here we note, interacting effects of elevated $CO_2$ concentrations and temperature or precipitation on biomass productivity (observed under experimental setups; e.g.

Obermeier et al. 2017) might obscure this attribution (Lombardozzi et al., 2018). Nevertheless, the assessment of the relative contribution as done by this approach seams valid as no significant interactions between these influencing factors on C stocks were observed within the TRENDY ensemble (Fernández-Martínez et al., 2019) nor within a fully coupled single model investigation (Devaraju et al., 2016).'

NEW (lines 247-249) – 'Note, by subtracting $f_{LULCC\_CO2}$ from the total $f_{LULCC}$ to derive the climate-only flux shares, this approach assumes zero synergies between effects of $CO_2$ concentrations and climatic changes on NBP in the DGVMs.'

Additionally, we apologize, as this confusion might have been caused by an error in our formula for $f_{LULCC\_Climate}$. The formula now reads $f_{LULCC\_Climate} = f_{LULCC\_trans} - f_{LULCC\_CO2}$ instead of $f_{LULCC\_Climate} = f_{LULCC\_Climate} - f_{LULCC\_CO2}$.

*Also, you don't report on the co2+no2 effects at all, so why is this simulation listed in the methods?*

We agree that this information was not necessary and removed it from Table 1 (columns *Nitrogen deposition* and *Nitrogen Fertilization*) and the text (lines 157-159). However, high spread between models might partly result from 'co2+no2' effects since some of the used model include nitrogen fertilization while others do not. We rephrased the respective wording in the appendix to highlight this as follows:

NEW (lines 534-538) - 'This pronounced model spread can be explained by intertwining issues, such as the low quality of historical LULCC data (with different data bases), the consideration or neglection of relevant processes (e.g. nitrogen fertilisation), the simplified representation and uncertainty in the parameterization of management and natural processes, uncertainties in soil and vegetation C stocks, and the lack of observational constraints (Friedlingstein et al., 2019, Gasser et al., 2020, Lienert et al., 2018, Goll et al., 2015, Li et al., 2017).'

*Results and discussion*

*lines 246-247:*

*unclear: "...to consistently use the same models for the flux and bias estimates on a spatio-temporal level..."*

We apologize for the confusion and changed the sentence also in line with your next comment '[...] to consistently use the same models where possible [...]' (line 263).

*line 254:*

*did you use the trendyv9 version of sdgvm throughout? if so, you need to state this in the methods.*

Yes, we used TRENDY v9 output from SDGVM throughout the study. According to your suggestion, we added a sentence to section 2. Data and Methods and refer to it in the results section as follows:

NEW (lines 138-140) - 'Note, for SDGVM, model output from TRENDY v9 was used due to erroneous merging of land cover and LULCC datasets in earlier versions that caused a C loss over the period ~1900-1970 mainly in semi-arid regions.'

NEW (lines 264-267) - 'Slight differences (<0.1 PgC yr$^{-1}$) between $f_{LULCC\_trans}$ derived in this study and the DGVM-derived GCB2019 estimates are attributable to the fact that we used only a subset (n=12) of the models analyzed within the GCB2019 (n=15), to consistently use the same models where possible. Additionally, the inclusion of TRENDY v9 model output for SDGVM (for reasons refer to Sect. 2) caused, for example, a lower LASC of 0.8 PgC yr$^{-1}$ for 2009-2018 in this study (0.84 PgC yr$^{-1}$ with two decimal places) as compared to consistently using TRENDY v8 output, where the resulting LASC (usually rounded to one decimal place) is 0.9 PgC yr$^{-1}$ due to rounding of 0.85 PgC yr$^{-1}$.'

*lines 256-257:*

*not necessarily. the single year pre-ind estimate is closer to bk estimates, and is equally close to gasser as the present-day for the decade.*

Thank you, we rephrased the sentence as follows:

NEW (lines 268-270) - 'As expected (Sect. 1), $f_{LULCC\_pd}$ is the DGVM-based $f_{LULCC}$ estimate that is most similar to the bookkeeping mean in the GCB2019 when compared over multiple years.'

*line 259:*

*you haven't shown the cumulative yet*

Thank you, we added a reference to the cumulative estimates in the beginning of the results section:

NEW (lines 260-261) - 'A general overview of most recent annual and cumulative estimates of $f_{LULCC}$ shows that our estimates are in good agreement to the published ones (Friedlingstein et al. 2019; Gasser et al. 2020; Tables 4 to 6).

*line 260:*

*awkward. try "...in the 1950's and again at the end of the simulation..."*

Thank you, we changed it according to your suggestion (line 278) .

*line 272:*

*"...comes into play..."*

Thank you, we changed it (line 300) .

*lines 272-273:*

*awkward first half of sentence.*

Thank you, we changed the wording of the respective sentence:

NEW (lines 300-303) - 'In general, beneficial environmental alterations for C sequestration widely increased the potential C stocks (Fig. 8), and thus, the LASC steadily increased (Fig. 2b,e), reaching about ~40% in recent annual and ~20% in cumulative contributions to $f_{LULCC\_trans}$ (Fig. 2c,f).'

*line 277:*

*during this period (first half of 20th century)*

Thank you, we changed the wording according to your suggestion:

NEW (lines 303-304) - '[...] still increase faster than the other estimates during this period (first half of 20th century; EED and PTD remain increasing) [...]'

*line 279:*

*"these peaks"*

Thank you, we change it (line 309).

*lines 281-284:*

*maybe because the LULCC occurrence is the same between simulations. but post-peak carbon stocks might have been lowered enough differentially between the sims to reduce the post-peak difference because carbon reductions are generally implemented as fractions, in which case the same LULCC would reduce more carbon in the pd sim than in the pi sim because of higher carbon stocks in the pd sim. and the eed peak is actually after the other peaks and stretched out. so is this really independent of the lulcc timing, as there is a peak in lulcc area that coincides with this emission peak.*

We agree that the wording was misleading. We changed it accordingly as follows:

NEW (lines 311-314) - 'The latter is highlighted by the simultaneous peak in EED which in essence is the intersection of LULCCs with the difference in standing biomass and actual soil C stocks due to altered environmental conditions over the last hundred years (under pre-industrial vs present-day environmental conditions; compare Fig. 2a with Fig. 2b and Sect. 2.2.2).'

*line 293:*

*post 1950*

Thank you, we added 'post 1950' (line 324).

*lines 296-320:*

*This section is difficult to follow, and it isn't entirely clear what the main point is, other than that LULCC dominates the EED pattern. But more importantly, LULCC seems to drive the fLULCC pattern also, since the EED and the fLULCC patterns also match. temporally, which is does not need a regional breakdown to be shown. And it would appear that the difference in biomass between Pi and Pd is a main driver of the corresponding difference in fLULCC. This boils down to biomass+LULCC controlling fLULCC estimates, with some regional variation in how this relationship contributes to the global estimates.*

*If the EED pattern is the main point, then take out the other regional plots (5 and 6), as figs 3 and 4 and 7 and 8 show this. The regional plots are difficult to interpret, and so the discussion is helpful, but it should be more clear how the regional patterns contribute to the global patterns, as figure 8 can show (and 7 can show magnitudes better is the scales are matched).*

We agree that this section (Sect. 3.2) was not to the point and changed it according to your suggestion. Additionally, we shifted Figures 5-7 into the appendix and created new Figures 5 and 6 with unified y-scales, to ease the comparability. The new section mainly discusses EED and the

environmental effects on estimates of $f_{LULCC\_pi}$ and $f_{LULCC\_pd}$ and ends with a passage shortly describing the higher complexity when comparing with $f_{LULCC\_trans}$ (seen e.g. by trend reversal in PTD) and a connecting sentence to the following sections where the positive and negative LASC estimates are discussed as the underlying drivers:

NEW (lines 338-359) - 'A large sensitivity of cumulative $f_{LULCC}$ towards choice of pre-industrial vs present-day environmental forcing is found in vast stretches across the globe: EED cumulated >8 PgC in the USA (mainly eastern parts), Brazil (mainly southern parts) and Southeast Asia, >5 PgC in Russia, China, Equatorial Africa, Southern Africa, and >2 PgC in Europe (mainly eastern parts), Southwest South America and South Asia from 1800 until 2018 (Figs. 10e and A12). Strikingly, the last decade saw the tropics to become more dominant in positive EED than other regions due to recent clearings (Figs. 5 and 10f). All these reflect particularly forested areas where LULCCs caused highest $f_{LULCC}$ quantities (Figs. 5 and 9a,c,e; compare increasing deviation of linear model from 1:1 line with higher values) due to the conversion of land with high NBP where positive changes in potential C stocks between 1800 and 2018 occurred (Fig. 8).

Conversely, very distinct regions of negative cumulative EED in Central Europe reflect relatively increased negative $f_{LULCC\_pd}$ due to early and widespread reforestation causing increased C uptake (Fig. 9e). Such a strong C uptake due to reforestation causes also globally wide-spread negative EED values in the last decade (Fig. 10f; with hotspots in northeastern Brazil, southern Africa and the Eurasian steppe zone), while the poor representation of recent large-scale reforestation programs with a concomitantly increased C sink in China (Lu et al., 2018; Chen et al., 2019) in the LUH2 data prevents EED (and also $f_{LULCC}$ estimates) to become negative in this region.

Regions, for example remote rain forests, that were only affected little by LULCCs hardly show up in EED. The pattern of EED is thus dominated by the pattern of LULCC with variations due to ecosystem sensitivity (namely in NBP) to environmental conditions (see Fig. 7). This shows that the choice of pre-industrial vs present-day environmental conditions can play a substantial role in regional $f_{LULCC}$ attribution.

As seen for the global estimates, the approach to derive $f_{LULCC}$ under transient environmental conditions introduces even more complexity, as it includes the LASC and strongly depends on the timing of LULCCs. In line, PTD undergoes a trend reversal with widely negative values in the most recent period in many regions (Figs. 6c,d and 10d), which we discuss in detail in the next section.'

[Figure]

Figure 5. Regionwise smoothed multi-model mean annual $f_{LULCC\_trans}$ (a&b), $f_{LULCC\_pi}$ (c&d), and $f_{LULCC\_pd}$ (e&f) from 1800 to 2018, derived according to Eqs. 1 to 3. For discussion on individual models refer to Sect. A1 and Figures A7 to A9. $f_{LULCC\_pd}$ was not derived for CLM5.0, JULES, LPJ and OCN (Table 2).

[Figure]

Figure 6. Regionwise smoothed multi-model mean annual Loss of Additional Sink Capacity (LASC; a&b), difference between fLULCC under present-day and pre-industrial environmental conditions (Environmental Equilibrium Difference, EED; c&d), and 'Present-day' vs 'Transient' environmental conditions Difference in $f_{LULCC}$ (PTD; e&f) from 1800 to 2018, derived according to Eqs. 4 to 6. For discussion on individual models refer to Sect. A1 and Figures A4 to A6 and A10 to A12. PTD was not derived for CLM5.0, JULES, LPJ and OCN (Table 2).

*Plots 9-11 are relevant, but they are not used to explain why the EED pattern dominates fLULCC, but they can support the basic relationship between biomass, LULCC trajectory, and fLULCC.*

*There are too many plots that are too difficult to read, and their referencing is difficult to follow also.*

We agree and changed the figures in the main manuscript also according to your previous comment and next comments. Thus, there is one less figure in the main part with the two new figures containing only 6 panels (compared to 16 in the earlier version of the regionwise plots). In line, we changed the referencing to new Figures 5 and 6 and rearranged the occurrence of the figures.

*lines 302-303:*

*The regional comparison plots (figures 5 - 7 and the supplemental) are difficult to compare because they are all on different scales. while it may make it more difficult to see some of the individual lines, putting them on the same scale is the best way to show the differences, which I think is the point.*

*You can split these plots into multiple figures also because they are so small, which also makes them difficult to read. If that is too many figures, then maybe all the regional plots should be supplemental.*

Thank you for these suggestions. Accordingly, and to ease the understanding of the main story, we created new Figures 5 and 6 to show regionwise multi-model mean emission estimates on unified y-scale for each hemisphere. We shifted all region-wise plots showing individual model performances into the appendix; individual model performances in the main script are now only found for global estimates.

*lines 321-352:*

*This section is difficult to follow, and it isn't clear what the point is of the comparison between LASC and EED. LASC is a component of fLULCC estimation, and EED is a difference between two estimates of fLULCC that don't include LASC. In general, the dynamics discussed here are still driven by biomass+LULLCC, since the forests are the high biomass areas and where LASC will have the most pronounced influence. While the regional breakdown and LASC discussion is somewhat informative, as the LASC dynamics are interesting and a key part of this study and since there is some regional variation, the relationships between LASC and EED seem to still be similar across regions, in that LASC tends to grow over time due to LULCC timing, while EED is driven mainly by biomass differences and the fixed LULCC trajectory. So is the point here to just discuss LASC dynamics, or to support the hypothesis that LASC is critical to fLULCC estimates because static biomass conditions do not include temporal effects of changing conditions (and drive errors associated with these biomass conditions)? I don't think you need the EED stuff here, as your larger point about estimation methods is discussed elsewhere (section 4?).*

It is true that LASC and EED are not directly related to each other. Nevertheless, we believe that a comparison of the two is helpful as the EED spans the 'realistic' range of possible $f_{LULCC}$ estimates that can be derived within the historic period. Thus, one could assume that the $f_{LULCC}$ estimates under transient simulations (where, in the end of the simulation, C densities approach the ones under present-day simulations) would never exceed the $f_{LULCC}$ estimates under present-day conditions. However, given the accumulating nature of the LASC, transient $f_{LULCC}$ is widely higher compared to present-day $f_{LULCC}$ in particular for most recent estimates, which by definition results in higher LASC values compared to EED (and consequently a negative PTD).

We clarified the purpose of Sect. 3.2.1 (and of the following) by rewording of Sect. 3.1 (see comment above) and restructured and modified the text in 3.2.1:

NEW (lines 370-394) - 'While EED is more relevant than the LASC for cumulative industrial-era emissions (change of sign in correlation in inlet Fig. 10e compared to Fig. 10f), the accumulating LASC heavily alters recent $f_{LULCC}$ estimates – Fig. 10b shows which regions would be attributed much higher emissions when the LASC is included in the $f_{LULCC}$ definition. Aggregated time series for the RECCAP2 regions reveal that the LASC started to increase ∼1850 in the USA, Russia and Southeast- and South Asia, ∼1900 in SW South- and Central- America and Southern Africa (Figs. 6a,b and A4). It then becomes even more pronounced ∼1950 in Brazil, Equatorial Africa and China, with the latter two and Southeast Asia showing a particular strong increase after 2000 (Figs. 6a,b, 10b and A4). Overall, the LASC accumulated to more than 4 PgC in the USA, Brazil, Equatorial Africa and Southeast Asia, and to 2–4 PgC in China, Russia, SW South- and Central- America, Southern Africa and South Asia (Figs. 10A and A10). As stated above, these high cumulative and annual LASC estimates mainly result from an initial high forest coverage and subsequent C losses in particular on areas where higher C stocks resulted from environmental changes over time (Sect. 3.4 and Fig. 8). Due to the different start of organized human agricultural, the forest clearings in the USA (mid of 19th century, with an early LASC initiation) though on forests with comparably low C stocks (Fig. A3) have caused similar cumulative sums as a much later onset of wide-spread LULCCs (beginning of 20th century) in Brazil, Equatorial Africa, China, and Southeast Asia (Fig. A10) due to rapidly increasing and pronounced higher vegetation C stocks in these regions (with strong response to $CO_2$ increase).

The widely negative PTD values across the globe for the period 2009–2018 indicate that the LASC causes recent $f_{LULCC}$ estimates from the current DGVM approach (under transient conditions) to be higher compared to bookkeeping estimates (which are similar to $f_{LULCC\_pd}$). However, small areas exist where EED remains larger than the LASC (i.e. positive PTD values) for the recent decade, here, more recent LULCCs caused even shorter period for the LASC to accumulate: in the tropics (mainly Brazil, Tanzania, Indonesia), sub-tropics (Eastern China, Southern Australia), and in the transition zones from temperate to boreal zone (Scandinavia, Russia). These regions would likely be attributed higher emissions by bookkeeping approaches than by $f_{LULCC\_trans}$ from DGVMs. This highlights another difficulty especially in regional $f_{LULCC}$ attribution: as the LASC accumulates emissions caused by past LULCCs, recent LULCCs are given less weight in relative terms. This also applies to recent LULCCs reducing atmospheric $CO_2$ such as reforestation, which cannot quickly compensate for past LULCC in approaches including the LASC, while they could in $f_{LULCC\_pi}$ and $f_{LULCC\_pd}$ estimates.'

*lines 353-377:*

*This discussion is a bit more clear as it focuses on LASC dynamics and doesn't mix in references to other calculations. I suggest combining this section and the previous section and focusing just on LASC dynamics. And the last sentence seems less a concluding sentence and more an evidence sentence to support the discussion.*

Due to the outstanding importance of the LASC for $f_{LULCC}$ estimation under transient environmental conditions, we kept the structure of the manuscript. In Sect. 3.2.1 we discuss the well-known LASC and associated lost C sinks, for the first time on a spatially explicit level, with the aim to highlight regions whose $f_{LULCC}$ estimates show strongest susceptibility to LASC inclusion and explain why the LASC can be greater than EED. In the following section (Sect. 3.2.2), we discuss the not so well-known occurrence of negative LASC values in some regions and particularly in most recent periods. To make this more clear we rephrased several parts (compare also earlier comments) and added following concluding sentence:

NEW (lines 420-423) - 'Comparing the different $f_{LULCC}$ estimates over time and across space, previous discussion has shown that the choice of method to derive $f_{LULCC}$ strongly impacts the estimated quantities. The effects of the interaction of the environmental forcing with the timing of the actual LULCC is particularly pronounced for estimates under transient environmental forcing and where NBP was strongly altered by environmental changes.'

*line 383-384:*

*What about LULCC effects due to abandonment of agricultural land? This also contributes to higher biomass in some places, and isn't trivial. You show some evidence of this in figure 12, where it can outweigh the climatic effects in even the long term, but your net results over the whole period do not show how regrowth has contributed more recently to increased biomass in places where this doesn't outweigh previous emissions.*

Indeed, some areas experienced LULCCs that were beneficial for the carbon stocks, such as abandonment of agricultural land. However, for the derivation of $CO_2$- vs climate-related $f_{LULCC}$ shares we used the total stock alterations of the S1 and S2 simulations, both of them have fixed pre-industrial LULCC forcing. Due to the fixed LULCC forcing, effects from LULCCs can be excluded, which we now explicitly state in the updated manuscript:

NEW (lines 426-428) - 'The mean simulated global vegetation C stock increased by ~23% from 664 PgC to 815 PgC from 1800 until today, in both the S1 and S2 simulation (which by protocol exclude effects from LULCCs; see Figs. 8 and A3 for maps and Fig. A1b for global estimates).'

*lines 416-445:*

*This section seems to be referring to a political question for emission attribution to regions for policy, rather than a scientific question of what are LULCC emissions.*

Thank you, we corrected the wording and highlighted the political context of this discussion as follows:

NEW (lines 465-469) - 'From a policy standpoint and disregarding considerations from the natural land sink perspective, these results highlight the need for a $f_{LULCC}$ estimate that is comparable over time and across space.'

NEW (lines 522-524) - 'Within the political context, if environmental effects on C sink capacity are to be accounted for in regional budgets (requiring DGVMs for the assessment) we argue for a consistent method which includes the LASC and emphasize that care must be taken in choosing the beginning of the accounting period.'

*So you should separate these two questions/recommendations. Is your scientific recommendation to discard some LULCC emissions? It doesn't seem to be so as you clearly show that LASC is an important component in "Accurate quantification of the net carbon flux from land use and land cover changes..." (line 447). Whether this makes sense for regional carbon policy is a different question.*

*From a policy standpoint, the question here is then why include only part of the total emissions from LULCC (i.e. LASC from second half of 21st century) while including some estimate of the directly effects of changes all the way back to 1850? If it isn't ok to include LASC back to 1850, why should any effects of pre-1950 LULCC be included? Why not just start full attribution of emissions at 1950?*

Thank you. We agree that the definition of the reference period only for the LASC was somehow arbitrary chosen. From a political standpoint, and in line with your suggestion, one could argue to derive not only the LASC but also $f_{LULCC}$ for such a unified reference period. Considering your comment and a comment of Reviewer #2, we now name different possibilities and carefully rephrased the whole paragraph as follows (in mind that this is a political decision which we leave open):

NEW (lines 479-498) - 'To circumvent these issues, as could be desired in the political context, one could derive $f_{LULCC}$ and the LASC based on simulations forced with the cycled climate and $CO_2$ conditions that occurred during the actual LULCC event. However, this would still result in differing accumulation periods and varying environmental conditions during and following a LULCC event. While the influence of the latter could be reduced using cycled pre-industrial or present-day environmental forcings, these neglect transient C stock changes. To consider the LASC but counteract spatial heterogeneity in $f_{LULCC}$ differences resulting from synergistic effects of environmental conditions and the timing of LULCCs, one could derive $f_{LULCC}$ and LASC from a defined reference period which is independent of the actual time that LULCCs occurred and shares the same reference conditions. For example, $f_{LULCC}$ and LASC could always be modelled for the second half of the 21st century, as here the environmental C stock changes have been amplified due to the accelerating increase of atmospheric $CO_2$ concentrations (alternative start times are of course conceivable). By using such reference period, the LASC could fully be captured also for most recent LULCCs (may they act positive or negative on C stocks) and foregone sinks would be more equally counted (same length of accumulation period with similar environmental changes). Along these lines, it may be considered to calculate such adapted LASC based on $CO_2$-only simulations as here the impact of humans is more homogeneously distributed, while the spatially heterogeneous climate impact on $f_{LULCC}$, determined foremost by action outside the location of LULCCs, causes a questionable attribution of regional $f_{LULCC}$ when compared across the globe (without even considering externalized $f_{LULCC}$ s e.g. due to remote market demand of food and timber; Lambin and Meyfroidt 2011; Meyfroidt et al., 2013). To detach $f_{LULCC}$ estimates from the timing of LULCCs and the spatially heterogeneous climate evolution, we argue to address the delineation of an adapted LASC in future studies, where, in particular, the reference to calculate the LASC should further be investigated. Such methodology could limit $f_{LULCC}$ to locally determined factors (namely LULCCs) and reduce the dependence on the timing of LULCCs while still reflecting the foregone C sink capacity by human intervention.'

*liens 467-472:*

*Here you clearly state that full accounting includes LASC. Then you vaguely backtrack and mention the derived LASC without context, which is the political context of attribution for policy. The "arbitrary length of their simulation period" is a given fact for all modeling and is handled by explicit definition of the period in question. For example, climate change is defined in reference to the "pre-industrial" period (usually pre-1850 equilibrium).*

Thank you, we apologize for the incorrect wording (in particular the misleading description of 'full accounting') and the missing link for the political context of attribution for policy. We changed the sentences as follows:

NEW (lines 522-527) – 'Within the political context, if environmental effects on C sink capacity are to be accounted for in regional budgets (requiring DGVMs for the assessment) we argue for a consistent method which includes the LASC and emphasize that care must be taken in choosing the beginning of the accounting period. As LASC values derived by the approach taken so far in the GCB are widely independent of locally determined environmental changes (rather depend on globally determined climatic changes) and strongly dependent on accumulation periods (defined by the timing of LULCCs and the end year of the simulations), we argue for a $f_{LULCC}$ attribution that is more robust against choices of environmental drivers and accumulation period by using an adapted LASC, for example, based on a defined common reference period and homogeneously altered environmental conditions (such as only driven by $CO_2$ alterations).'

*Figures*

*Your figures and supplemental figures are out of order with respect to references in the text.*

We changed the order of the figures in the main text by adjusting the appearance and order of tables (Table 1 is changed with Table 2) and figures (shifting Figures 5-7 into the appendix, and creating new Figures 5 & 6). Additionally, we added additional references in the text at several places (compare track changes document) and put additional labels in the figures to ease comparability.

*Consider using the same scale across the regional comparison plots, both in the paper and in the supplemental.*

Thank you for this suggestion. As highlighted before, we created new Figures 5 and 6 showing region-wise multi-model mean emission estimates on unified y-scale for each hemisphere. However, since the emissions quantities of the different regions are extremely different (> factor of ten variation), a clear depiction of individual models on the same scale for each region is not feasible. Therefore, we kept the region-wise plots showing individual model performances on individual scales, but now assemble all of them in the appendix section.

*Appendix A1*

*lines 484-485:*

*This statement appears incorrect based on the discussion that follows.*

*It appears that the variation may arise from models with different ecological and productivity responses to the cycled environmental conditions.*

Thank you, we shifted the respective statement into Sect. 3.1 and rephrased it (in line with a comment from Reviewer #2) as follows:

NEW (lines 283-288) - 'Note, a high internal climate variability translated into a high interannual variability in NBP and consequently a high variability of $f_{LULCC}$ estimates (Figs. 3, 5 and A13) and of their respective differences (Figs. 4 and 6). For the differences in $f_{LULCC}$ estimates, some artefact might additionally arise due to comparison of simulations with different forcing cycles (e.g. on global scale, with periodic fluctuations in annual relative shares of EED to $f_{LULCC\_pd}$ in Fig. 2c and, on regional scale, in Figs. 6 and A4 to A6 with pronounced oscillations in some regions).'

######################################################################

In addition to changes resulting from the reviewer comments, we made the following changes:

- changed to capitalized journal abbreviations in references and added DOI that where missing
- added new Table 3 with overview of different estimates
- unified usage of text styles for abbreviations (italic vs normal) in equations and text

######################################################################

New references:

Devaraju, N., Bala, G., Caldeira, K., and Nemani, R.: A model based investigation of the relative importance of $CO_2$-fertilization, climate warming, nitrogen deposition and land use change on the global terrestrial carbon uptake in the historical period, CLIM DYNAM, 47, 173–190, https://doi.org/10.1007/s00382-015-2830-8, 2016.

Fernández-Martínez, M., Sardans, J., Chevallier, F., Ciais, P., Obersteiner, M., Vicca, S., Canadell, J., Bastos, A., Friedlingstein, P., Sitch, S., et al.: Global trends in carbon sinks and their relationships with $CO_2$ and temperature, NAT CLIM CHANGE, 9, 73–79, https://doi.org/10.1038/s41558-018-0367-7, 2019.

Hegerl, G. C., Brönnimann, S., Cowan, T., Friedman, A. R., Hawkins, E., Iles, C., Müller, W., Schurer, A., and Undorf, S.: Causes of climate change over the historical record, ENVIRON RES LETT, 14, 123 006, https://doi.org/10.1088/1748-9326/ab4557, 2019.

Lombardozzi, D. L., Bonan, G. B., Levis, S., and Lawrence, D. M.: Changes in Wood Biomass and Crop Yields in Response to Projected $CO_2$, $O_3$ , Nitrogen Deposition, and Climate, J GEOPHYS RES-BIOGEO, 123, 3262–3282, https://doi.org/10.1029/2018JG004680, 2018.

Obermeier, W. A., Lehnert, L. W., Kammann, C., Müller, C., Grünhage, L., Luterbacher, J., Erbs, M., Moser, G., Seibert, R., Yuan, N., et al.: Reduced $CO_2$ fertilization effect in temperate C3 grasslands under more extreme weather conditions, NAT CLIM CHANGE, 7, 137–141, https://doi.org/10.1038/nclimate3191, 2017.